# Systems acclimation to osmotic stress in zygnematophyte cells

Jaccoline M. S. Zegers [1] ✉, Lukas Pfeifer [2], Tatyana Darienko[1,3], Kerstin Schmitt [4,5], Craig A. Dorsch [1], Cäcilia F. Kunz [1], Gerhard H. Braus [4,5], Birgit Classen [2], Oliver Valerius [4,5] & Jan de Vries [1,6,7] ✉

Zygnematophytes are the closest algal relatives of land plants. They hold key information to infer how the earliest land plants overcame the barrage of terrestrial stressors, the prime of which is osmotic stress. Here, we apply two osmotic stressors on a unicellular and a multicellular representative of zygnematophytes and study their responses over a 25-hour time course, generating 130, 60, and 30 transcriptomic, proteomic, and metabolomic samples combined with photophysiology, sugar analysis, immunocytochemical glycoprotein analysis, and microscopy. Our data highlight a shared protein chassis that shows divergent responses with the same outcome: successful acclimation to osmotic challenges. We establish a model of how the algal sisters of land plants can overcome a prime stressor in the terrestrial habitat and highlight components of the plant terrestrialization toolkit.

Phylogenomic research over the last decade has consistently recovered the Zygnematophyceae as the closest algal relatives of embryophytes (land plants)[1–5]. Zygnematophyceae diversified into a species-rich class of unicellular and filamentous algae that live in diverse habitats[6,7]. Studying their genomes[8–11] revealed a whole set of genes that are homologous to the molecular chassis known from environmental adaptation in land plants[12–15]. Previous studies particularly focused on signaling cascades regulating stress responses[15–20]. Yet, our knowledge on the proteins that constitute the stress physiology of zygnematophyte cells is very limited. In this study, we focused on the protective molecular mechanisms against hyperosmotic stress, a prime challenge on land.

A defining characteristic of terrestrial habitats compared with aquatic environments is the much more variable availability of water. These fluctuations are closely linked to frequent osmotic stress: a decrease in water availability results in hyperosmotic stress, whereas an increase can lead to hypo-osmotic stress. Land plants have adapted to these challenges through diverse strategies for managing changes in water availability. They have evolved specialized adaptations, including water-conducting tissues in vascular plants and water-conducting cells in bryophytes[21–23]. In addition, stomata[24] and the cuticle[25–27] are characteristic features of embryophytes, serving to regulate water loss and to retain water within the plant, respectively.

At the cellular level, water regulation is vital for maintaining structural integrity and function. Plant and algal cells possess cell walls, onto which intracellular fluid mediates an outward force known as turgor pressure, critical for maintaining cell structure and function. Under hyperosmotic stress—such as when osmolyte concentrations in the surrounding environment increase due to water loss—turgor pressure decreases as water movement into the cell diminishes or reverses. This pressure drop triggers a cascade of physiological and

[1]University of Göttingen, Institute for Microbiology and Genetics, Department of Applied Bioinformatics, Göttingen, Germany. [2]Christian-Albrechts-University of Kiel, Pharmaceutical Institute, Department of Pharmaceutical Biology, Kiel, Germany. [3]University of Göttingen, Albrecht-von-Haller-Institute for Plant Sciences, Experimental Phycology and Culture Collection of Algae at Göttingen, Göttingen, Germany. [4]University of Göttingen, Institute of Microbiology and Genetics, Department of Molecular Microbiology and Genetics, Göttingen, Germany. [5]University of Göttingen, Göttingen Center for Molecular Biosciences (GZMB), Service Unit LCMS Protein Analytics, Göttingen, Germany. [6]University of Göttingen, Göttingen Center for Molecular Biosciences (GZMB), Department of Applied Bioinformatics, Göttingen, Germany. [7]University of Göttingen, Campus Institute Data Science (CIDAS), Department of Applied Bioinformatics, Göttingen, Germany. ✉e-mail: jaccoline.zegers@uni-goettingen.de; devries.jan@uni-goettingen.de

biochemical responses, including structural alterations, membrane leakage, molecular crowding, and other forms of cellular damage[28,29].

To mitigate the effects of hyperosmotic stress, plants have evolved several adaptive mechanisms. A palpable response well-known from tracheophytes is the closure of stomata to minimize water loss[30,31]. On the cellular level, plants and algae initiate membrane reorganization and synthesize osmotically active metabolites, such as proline, sugars, and sugar alcohols, to stabilize the intracellular osmotic potential[32]. When dehydration results from osmotic stress, some algae can down-regulate the water-intensive process that is photosynthesis[33]. Reactive oxygen species (ROS) are also commonly produced during drought-related stresses, being both signals and culprits emerging from an imbalance in photosystems[31,34–36]. In response, plants activate ROS-scavenging systems to prevent oxidative damage; plants further produce a variety of stress-responsive proteins that play essential roles in protecting cellular membranes, maintaining ionic homeostasis, and degrading damaged proteins to facilitate repair[28,37]. Conserved structural modifications to the cell wall and cytoskeleton[38] bolster cellular strength and resilience to the mechanical stresses associated with turgor pressure loss. Collectively, these mechanisms represent a multifaceted strategy that allows plants to survive and adapt to osmotic stress conditions. These key mechanisms of osmotic stress tolerance have been studied mainly in land plants, leaving their evolutionary conservation obscure.

Some zygnematophyceaen algae exhibit exceptional adaptability to osmotic stress, thriving in habitats characterized by frequent fluctuations in water availability, including ponds, puddles, and Arctic regions[39–41]. Previous studies, often with a focus on one specific technical aspect, have identified key coping strategies and hyperosmotic stress-responsive genes in these algae; these include upregulated glycosyltransferase family 2 genes in *Zygnema circumcarinatum*[42] and microscopic observations of cell wall and endomembrane adaptations in *Penium margaritaceum*[43]. Here, we employ a systems biology approach of temporal transcriptomics and proteomics to uncover the core osmotic stress response in Zygnematophyceae. From our data, we infer key components of the plant terrestrialization toolkit.

## Results

### Quantitative impacts of osmotic stress in *Mesotaenium* and *Zygnema*

We worked with two representatives of the closest algal relatives of land plants: the unicellular *Mesotaenium endlicherianum* SAG12.97 and the filamentous *Zygnema circumcarinatum* SAG698-1b, hereafter referred to as *Mesotaenium* and *Zygnema*. Next to representing the two predominant body plans in zygnematophytes, these species were selected due to (i) their natural exposure to osmotic stress in their habitats: *Mesotaenium* originates from a lake subject to annual drying cycles, while *Zygnema* is found in a meadow ditch; (ii) the availability of fully sequenced genomes and updated gene models[8,10,15]; and (iii) the presence of all protein components involved in the canonical abscisic acid (ABA) signaling cascade[10,44], critical in osmotic stress response of embryophytes[45,46]−albeit these proteins likely act independent of ABA in zygnematophytes[18].

*Mesotaenium* and *Zygnema* were cultivated on cellophane disks placed on agarised medium, allowing transfer to other agar plates without disturbing the algae (Fig. 1a). Unlike cultivation in liquid medium, this approach allowed for stress experiments that more closely mimic terrestrial habitats while enabling easy and rapid sampling. The osmotic stress treatments for this study were initially based on Kilian et al.[47]: 300 mM mannitol and 150 mM NaCl for 24 hours. However, during preliminary trials, no clear morphological response was observed under 300 mM mannitol within 24 hours. Consequently, the mannitol concentration was increased to 800 mM. In contrast, treatment with 150 mM NaCl (both osmotic and ionic stress) led to a clear and rapid drop in the maximum quantum yield of photosystem II, Fv/

Fm, that is a non-invasive and comparable measure of gross physiology[48].

The highest Fv/Fm values were observed 25 hours after transferring the algae to a new growth plate without osmolytes, likely reflecting optimal conditions provided by the fresh medium. In this treatment, Fv/Fm values ranged from $0.65 \pm 0.05$ for *Mesotaenium* and $0.73 \pm 0.02$ for *Zygnema* (Fig. 1b). Following the onset of osmotic stress, Fv/Fm values decreased in both algae ($P < 0.01$ for both treatments after 6 h, Fig. 1b). However, recovery patterns differed: *Zygnema* showed a rapid rebound in the NaCl treatment, whereas *Mesotaenium* exhibited a slower recovery. In contrast, Fv/Fm values remained $8.8\% \pm 5.1\%$ and $25.2\% \pm 6.1\%$ below their starting values under mannitol treatment for *Zygnema* and *Mesotaenium*,; they never approached zero, indicating that the algae retained some potential for photosynthesis.

Water moves from regions of higher concentration to regions of lower concentration; upon hyperosmotic stress, we expected an efflux and/or a reduced influx into the cell. The water content in both species was significantly ($P < 0.001$) reduced upon NaCl and mannitol treatments (Fig. 1c). Notably, the dry weight of the mannitol-treated samples was substantially higher compared to the NaCl-treated and control samples, but this is explained by an accumulation of mannitol in the samples as later confirmed through metabolite fingerprinting (see below).

A rapid decrease of water content triggers plasmolysis in plants: the movement of water out of the cell decreases the volume of the protoplast, which could detach from the rigid structure of the cell wall as also previously well-documented for *Zygnema circumcarinatum*[49]. Although plasmolysis was not complete in all cells, with the plasma membrane remaining connected to the cell wall in most areas, it was the most significant feature observed during the morphological analysis of both *Mesotaenium* ($P = 0.002$) and *Zygnema* ($P = 2.8e\text{-}5$) under high osmotic stress induced by mannitol treatment (Fig. 1c, 1d). In addition to plasmolysis, other structural changes included distortions of the chloroplast, bending and breaking of cells/filaments, changes in vacuolar structures and other inclusions (Figs. 1c, 1d). In *Mesotaenium* "cups" were formed, which are non-green and often reddish cell-wall-enclosed compartments that can detach from the mother cell.

### Dynamics in stress-responsive proteins and transcripts

To investigate the molecular response of the algae, we applied RNA sequencing at t0, 3 h, 6 h, 9 h, and 25 h and shotgun proteomics at t0, 6 h, and 25 h. Accurate transcript levels were determined for 21,651 transcripts in *Mesotaenium* and 9631 transcripts in *Zygnema*, with the number of analyzed genes being 28% higher for Mesotaenium and 2% higher for Zygnema compared to those analyzed in Rieseberg et al.[14]. Protein levels were reliably quantified for 2171 and 1713 proteins, respectively, 88% and 92% of which were detected in 100% of samples for at least one treatment, enabling their inclusion in differential abundance analyses. Further, a de novo transcriptome was assembled as a complementary database for proteomics analysis, yielding 90 and 200 additional protein groups in *Mesotaenium* and *Zygnema*, respectively. In *Mesotaenium*, many of these newly identified proteins corresponded to isoforms of incomplete genes located at the ends of contigs, while in *Zygnema*, they were predominantly new genes already annotated in *Zygnema circumcarinatum* UTEX 1559 (Supplementary Fig. 1 and Supplementary Data 1).

Principal component analysis (PCA, Fig. 2a) revealed that proteins from different treatments were better separated at the 25-hour timepoint compared to the 6-hour timepoint. The PCA of the transcript data provided further insights, revealing distinct differences in the speed and variation of responses between treatments between *Mesotaenium* and *Zygnema*: In *Mesotaenium*, controls showed minimal variation across timepoints, whereas *Zygnema* exhibited clear shifts along the second principal component over time. Each treatment

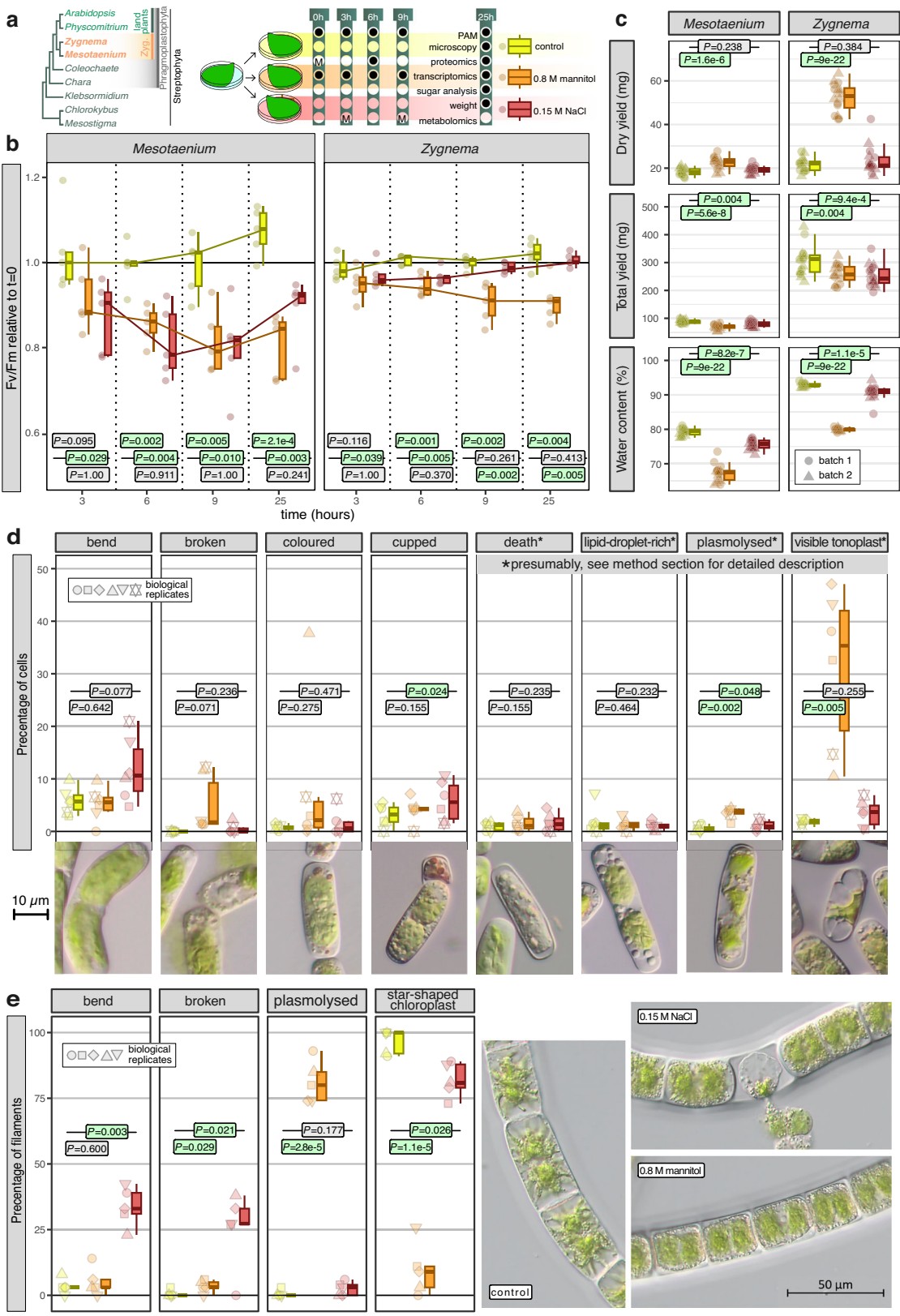

displayed unique temporal patterns: in *Mesotaenium*, early NaCl-treated samples were farthest from controls but later converged, indicating a rapid and strong response. In contrast, the transcript PCA for *Mesotaenium* revealed greater variability among biological replicates in the latest timepoints of the mannitol treatment, with some samples clustering near the controls while others diverged

substantially, suggesting an acclimation process that builds on large transcriptomic changes and bears out in a different temporal dynamic in the two species.

We performed a differential abundance analysis, comparing each treatment to the control for all timepoints (Fig. 2b, 2c). Consistent with the PCA results, protein fold changes were more pronounced at 25 h

**Fig. 1 | Physiological impact of osmotic stress in two zygnematophytes.**
**a** Cladogram of the Streptophyta, highlighting the two zygnematophytes (Zyg.) used and schematic overview of the experimental setup, depicting with a dot when, during the stress experiment, samples were taken for certain types of analyses. "M" stands for samples/analyses solely collected and performed for *Mesotaenium*.
**b** Relative photosynthetic efficiency (Fv/Fm) estimated from chlorophyll fluorescence measurements. *P*-values were calculated with pairwise multiple comparison tests (two-sided), based on Tukey contrasts with *p*-value Bonferroni adjustment. Each data point (see jitter) represents a biological replicate. **c** Weight per plate and water content measured after 25 hours of osmotic stress. *P*-values were calculated with pairwise multiple comparison tests (two-sided), based on Dunnet contrasts.

Microscopic features observed under light microscopy after 25 hours of osmotic stress in *Mesotaenium* (**d**) and *Zygnema* (**e**); *P*-values were calculated with paired one-sided t-Test. Boxplot colors (left to right): yellow = control, orange = 0.8 M mannitol, red = 0.15 M NaCl. **a–e** all *P*-values below 0.05 are highlighted in green. In all box plots (**b–e**), the box shows the interquartile range (IQR) from the 25th (Q1) to the 75th percentile (Q3) while the whiskers extend to minimum and maximum values in the data determined by Q1−1.5 × IQR to Q3 + 1.5 × IQR (data points outside are defined as outliers); the center line represents the mean. Different symbols of the jitter in (c) represent different batches and multiple occurrences of the same symbol indicate biological replicates (2x $n$ = 8); in (**d**) and (**e**) they represent biological replicates. Source data are provided as a Source Data file.

than at 6 h, whereas transcript levels began to change earlier, indicating a faster transcriptomic response compared to the proteomic response. The latter was most apparent in the salt treatment, which showed the strongest transcriptional changes between 3 and 9 hours, followed by a partial recovery—resembling the control more closely—after 25 hours. This pattern was especially clear in *Mesotaenium*, where the control timepoints showed minimal variation among themselves, and the final salt-treatment timepoint appeared more similar to the controls than to the earlier treatment stages. An intriguing difference was observed between *Zygnema* and *Mesotaenium*: in *Mesotaenium*, the fold changes for mannitol and NaCl treatments were similar, whereas in *Zygnema*, the mannitol treatment exhibited more pronounced fold changes than NaCl. These trends were consistent across both transcriptomic and proteomic levels.

Several proteins changed significantly in differential abundance (151 proteins summed in all analysis in both timepoints and both species; FDR < 0.05; Supplementary Data 2). Focusing on those identified in both *Mesotaenium* and *Zygnema*, five types were found. Among these, SVR3, a putative chloroplast TypA translation elongation GTPase, was the only protein downregulated. SVR3 has been implicated in chloroplast development under cold stress in plants[50], but its broader functional roles remain largely unexplored. Among the upregulated proteins, one of the most differentially abundant during osmotic stress was an NADPH-dependent aldo-keto reductase (AKR). AKR catalyzes the reduction of carbonyl groups to primary and secondary alcohols. It is frequently associated with abiotic stress responses, functioning in detoxification, osmolyte production, or the synthesis of secondary metabolites[51]. The remaining three upregulated proteins are predicted to localize in the cell wall: an exordium-like protein (EXL), xyloglucan endotransglucosylase/hydrolase (XTH), and proteins containing at least one fasciclin-like (FAS1) domain. XTH likely contributes to cell wall remodeling by modifying xyloglucan, whereas the functions of the other cell wall-associated proteins are less well understood. The number of transcripts showing significant changes is substantial (>4000 with $P_{adj}$ < 0.01), making it impractical to analyze all of them individually (see Supplementary Data 3). Focusing on the top five differentially expressed transcripts, we observe that both *Zygnema* and *Mesotaenium* share a common upregulated gene encoding a LEA (Late Embryogenesis Abundant) protein. LEA proteins play a crucial protective role during environmental stress in plants and are frequently upregulated in response to osmotic-related stressors across various species[52].

### Integrated analysis of transcript and protein correlations

Using transcriptome and proteome samples from the same plate, our parallelized experimental setup enabled robust comparisons that exposed discrepancies possibly attributable to translational control, post translation modification, or protein turnover. When examining how osmotic-stress-induced changes in specific transcript–protein pairs correlate (Figs. 3a, 3c, and Supplementary Fig. 2), we observe a strong correlation in *Mesotaenium* (R = 0.63) and a moderate correlation in *Zygnema* (R = 0.39). The correlations improve on average by 1.8x when using 25 h protein samples compared to 6 h samples, a

difference that exceeds the variability observed across transcript timepoints (maximum 1.2x). In this study, conserved stress responders such as AKR (both species), ELIP (*Mesotaenium*), and heat shock proteins (HSP, *Zygnema*) exhibited the strongest correlations, suggesting that their increased abundances result primarily from direct translation of elevated transcripts. For both species, several FAS1 proteins displayed a reversed pattern: higher protein abundance under hyperosmotic stress conditions but lower transcript levels. This pattern was also observed for the photosystem II subunit S (PsbS, Mesotaenium), ribosomal protein S5B (RPS5B, *Zygnema*) and expansin (*Zygnema*), suggesting the involvement of post-transcriptional regulatory mechanisms.

Examining the same data from a different perspective, we compared absolute transcript and protein levels (Figs. 3b, 3d, and Supplementary Fig. 2), finding a moderate correlation for *Zygnema* (R ≥ 0.34), except under mannitol treatment at 25 h, which suggests the potential involvement of post-transcriptional regulatory mechanisms. For *Mesotaenium*, the initial correlation between absolute protein and transcript levels was overall poor (R ≤ 0.28). Upon further examination, we found that many proteins with high abundance but unexpectedly low transcript levels are encoded on extremely small contigs (<4096 bp) or the plastome. Excluding genes located on small contigs significantly improves the correlation (*P* = 0.00026), surpassing that observed in *Zygnema*. In the following, we worked with integrating the consistent changes for specific protein-coding genes and proteins, detected via RNAseq and proteomics, to glean the molecular cell biological mechanisms that zygnematophytes use for acclimation to osmotic challenge.

### Temporal clustering unveils conserved osmotic stress responses

To delve deeper into the biological significance of the proteomic and transcriptomic datasets, we first divided the proteomic dataset into quadrants (Fig. 4a, Methods). Functional enrichment analysis yielded results consistent with the manual annotation performed for the statistically relevant genes. To cluster the substantially larger transcriptomic dataset we grouped temporal patterns for each transcript while retaining the distinctions between the treatment conditions. Temporal patterns were first grouped using a Gaussian Mixture Model (GMM) clustering; groups were normalized and further clustered using the same algorithm to reduce complexity. This approach resulted in 12 and 9 temporal modules for *Mesotaenium* and *Zygnema*, respectively, which were then subjected to functional enrichment analysis (Fig. 4b). In *Mesotaenium*, most modules exhibited a control pattern that remained consistently flat. Conversely, in *Zygnema*, transcript levels in the control treatment fluctuated markedly over time. Interestingly, in *Zygnema*, the temporal patterns of the salt treatment often appeared intermediate between those of the control and mannitol treatments. For *Mesotaenium*, the largest modules displayed similar response patterns for both mannitol and salt treatments, albeit with slightly faster responses observed in the salt treatment. Yet, the smaller modules also revealed treatment-specific responses unique to the mannitol and salt conditions.

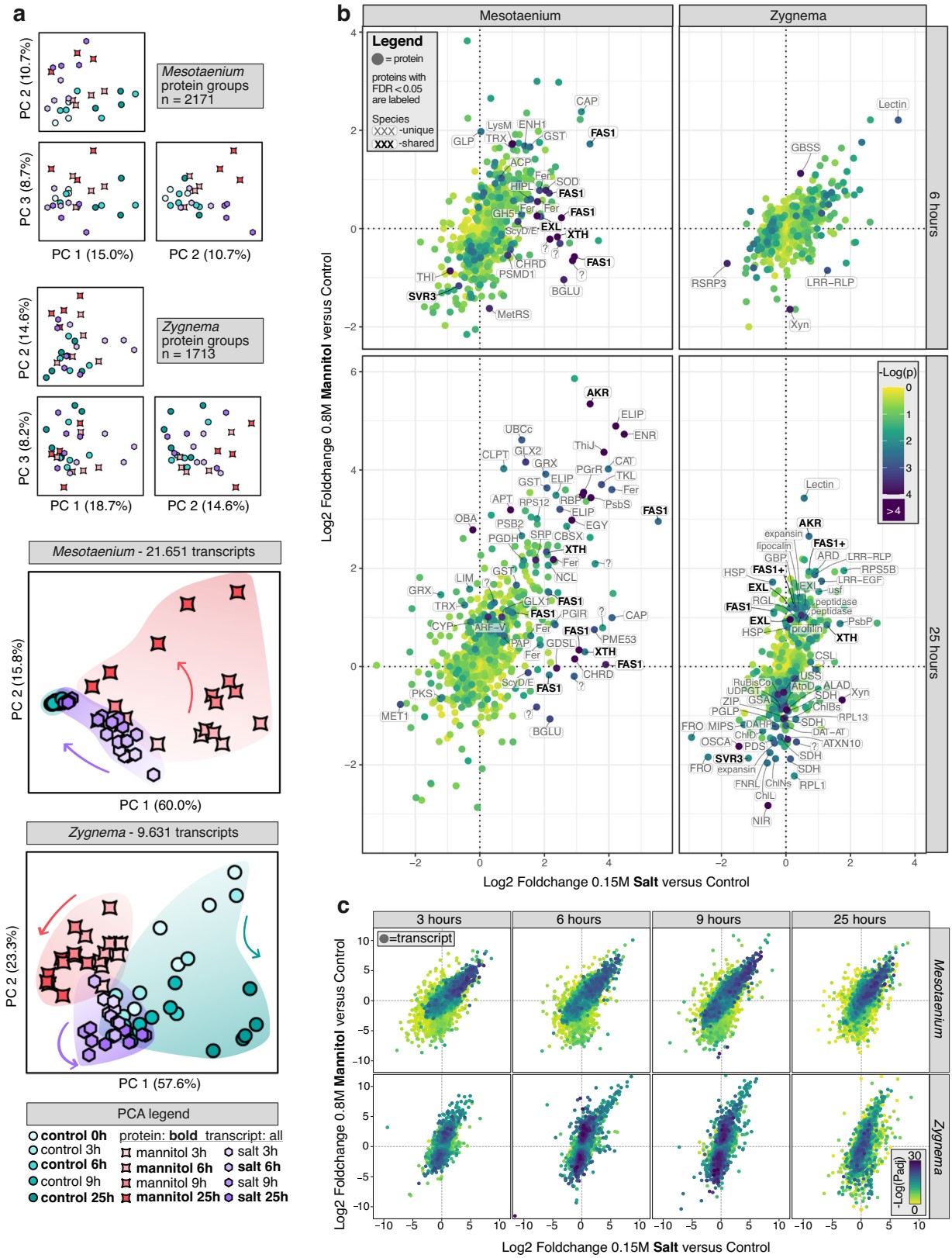

A total of 670 and 320 enriched terms ($p < 0.01$) were identified for *Mesotaenium* and *Zygnema*, respectively. Intersection network analysis (Fig. 5) revealed several highly connected terms, including oxidoreductase activity, membrane, and chlorophyll binding. These terms are functionally broad and are represented by both upregulated and downregulated proteins under osmotic stress. For instance,

"chlorophyll binding" includes both Early Light-Induced Proteins (ELIPs), commonly stress-inducible[53], and classical Light-Harvesting Complexes (LHCs).

The two most interconnected modules are Module 9 of *Mesotaenium* and Module 7 of *Zygnema*. Both modules exhibit an upregulated pattern under osmotic stress (Fig. 4b), suggesting a

**Fig. 2 | Differential proteome and transcriptome analysis in *Mesotaenium* and *Zygnema*. a** Principal component analysis (PCA) plots of log2-transformed label-free quantification (LFQ) values from the proteomics dataset and transcripts per million (TPM) values from the RNA-seq dataset. **b** Log2 fold changes in protein abundance for the 0.8 M mannitol treatment relative to the control, plotted against those for the 0.15 M NaCl treatment relative to the control. The proteins in bold are: AKR = NADPH-dependent Aldo-Keto Reductase, EXL = Exordium-like protein, FAS1 = protein with at least one Fasciclin-like 1 domain, SVR3 = putative chloroplast TypA

translation elongation GTPase, XTH = Xyloglucan endotransglucosylase/hydrolase. The full names of the proteins can be found in supplementary data 2. Log(P values) are based on multiple sample test ANOVA with a permutation-based FDR set to 0.05 and S0 set to 0.15. **c** Log2 fold changes in transcript abundance for the 0.8 M mannitol treatment relative to the control, plotted against those for the 0.15 M NaCl treatment relative to the control. Log(P-values) are based on scipy.stats multiple sample test with Benjamini-Hochberg FDR correction, independent samples, and an alpha set to 0.01.

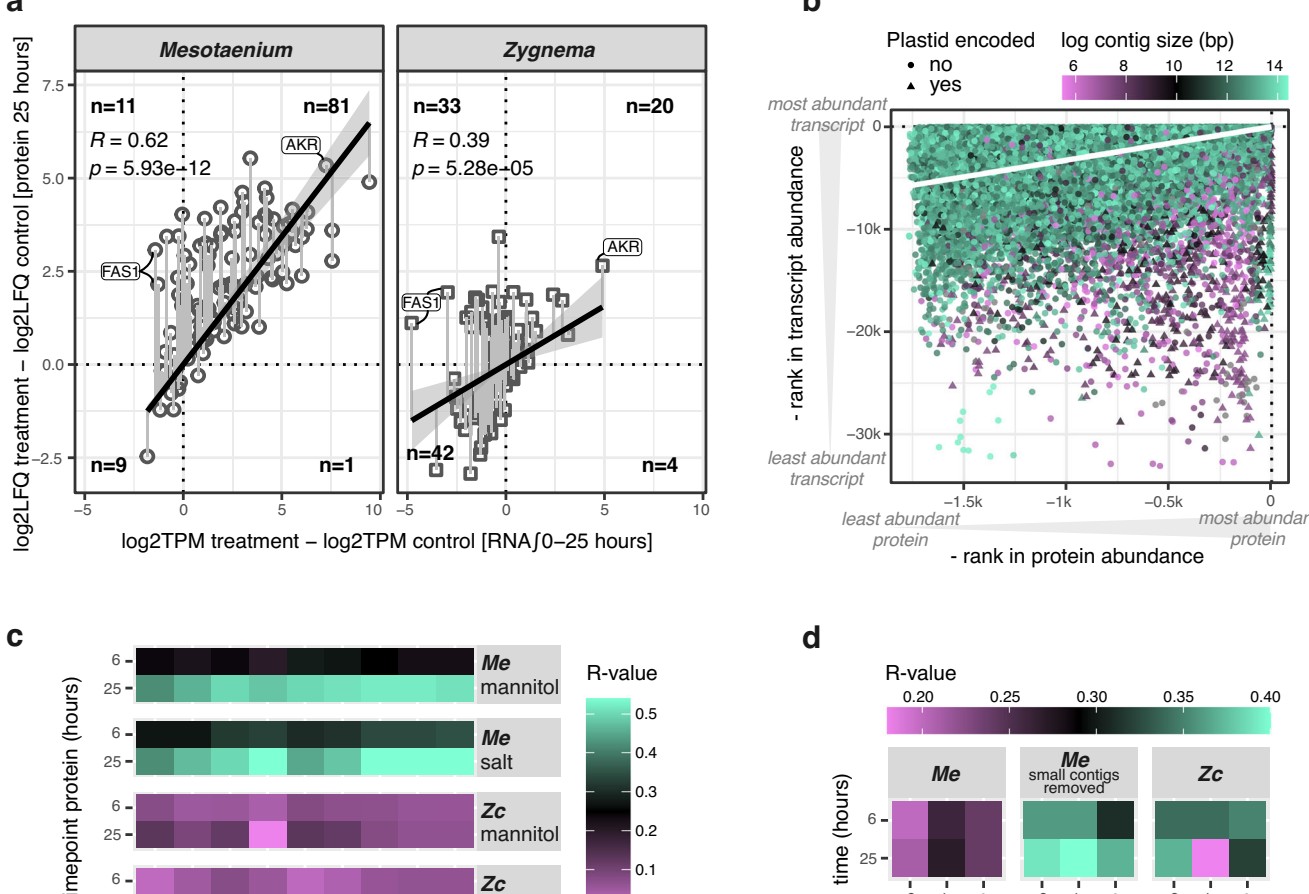

**Fig. 3 | Relationship between RNA and protein abundance during osmotic stress. a** Relative protein abundance at 25 hours compared to relative RNA abundance (integrated from 0 to 25 hours) for proteins showing a significant fold change (FDR < 0.05) in either salt or mannitol treatments relative to the control. The vertical lines from the trendline to the data points represent residuals. *P*-values on correlations were calculated based on stat_corr t-test. **b** Absolute protein abundance in *Mesotaenium* at 0, 6, and 25 hours plotted against absolute RNA levels at corresponding time points and replicates, with data points colored according to

the size of the contig from which the protein-coding gene originates. The negative rank indicates the relative abundance of each protein, with −1 representing the most abundant protein, −100 the 100th most abundant, and so on. The white line represents the linear trendline. **c** Pearson correlation coefficients between relative (log2 fold change) protein abundance at 6 or 25 hours and corresponding relative RNA abundance. **d** Pearson correlation coefficients between absolute protein and RNA abundance at 6 or 25 hours.

conserved osmotic stress response within the Zygnematophyceae. The uniquely shared terms within these modules are represented in both *Mesotaenium* and *Zygnema* by autoinhibited plasma membrane proton (H⁺) ATPases (AHA), sucrose synthase (SuSy), tonoplast intrinsic protein (TIP), and synaptotagmin (SYT). The AHA proteins are responsible for generating an electrochemical proton gradient across the plasma membrane, a crucial process for pH regulation and transport across the membrane[54]. Sucrose synthase catalyzes the reversible reaction converting sucrose into fructose and UDP-

glucose, playing a vital role in energy metabolism, osmolyte production, and complex carbohydrate biosynthesis[55]. TIPs are aquaporins located in the tonoplast—the vacuolar membrane—that facilitate the transport of water and small solutes across this membrane[56]. SYTs are membrane-associated proteins with an N-terminal domain located in a membrane, e.g. the *At*SYT1 is located in the ER-membrane; their C-terminal domain binds to other membranes, enabling them to bring membranes together, organize membrane structure, and facilitate lipid transfer[57].

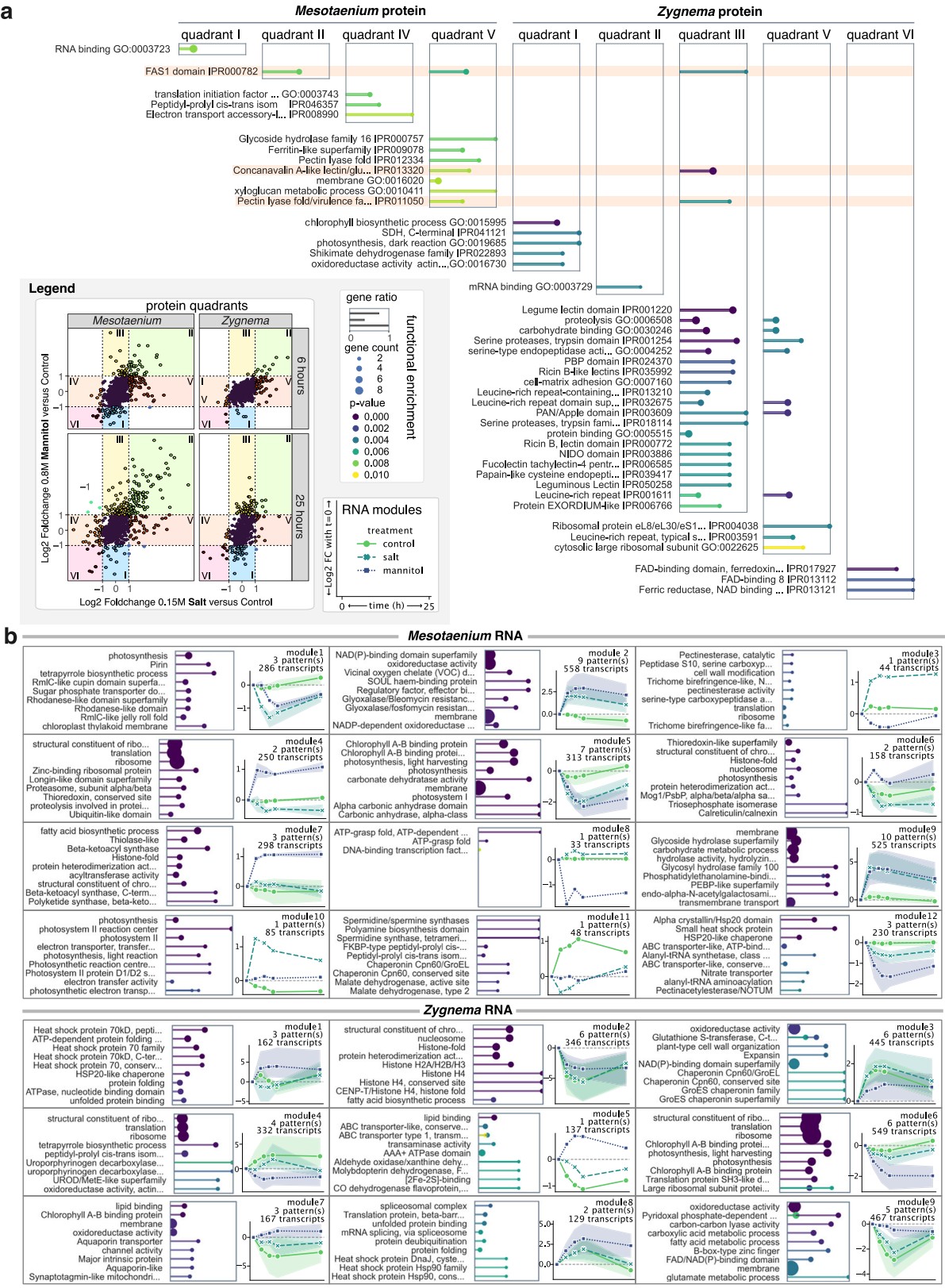

**Fig. 4 | Functions enriched in two zyngematophytes challenged with osmotic stress.** Functional Enrichment of GO Terms and InterPro Domains During Osmotic Stress (FDR < 0.01). **a** Proteins were grouped into quadrants based on log2 fold changes (≥1 or ≤-1), highlighting significant changes in response to 0.15 M NaCl and 0.8 M mannitol treatments. Functional enrichment analysis was performed on each group to identify enriched GO terms and InterPro domains. The three terms enriched in both *Mesotaenium* and *Zygnema* are highlighted in orange. The *P* values

were calculated by Fisher's exact test. **b** Transcript temporal changes were clustered into patterns using a Gaussian Mixture Model, with normalized patterns further organized into modules. Functional enrichment analysis identified enriched terms for each module, with the top nine displayed. The colored shading shows the standard deviation of the unnormalized gene expression patterns that make up the modules. The center (symbol) in the error bands represents the mean.

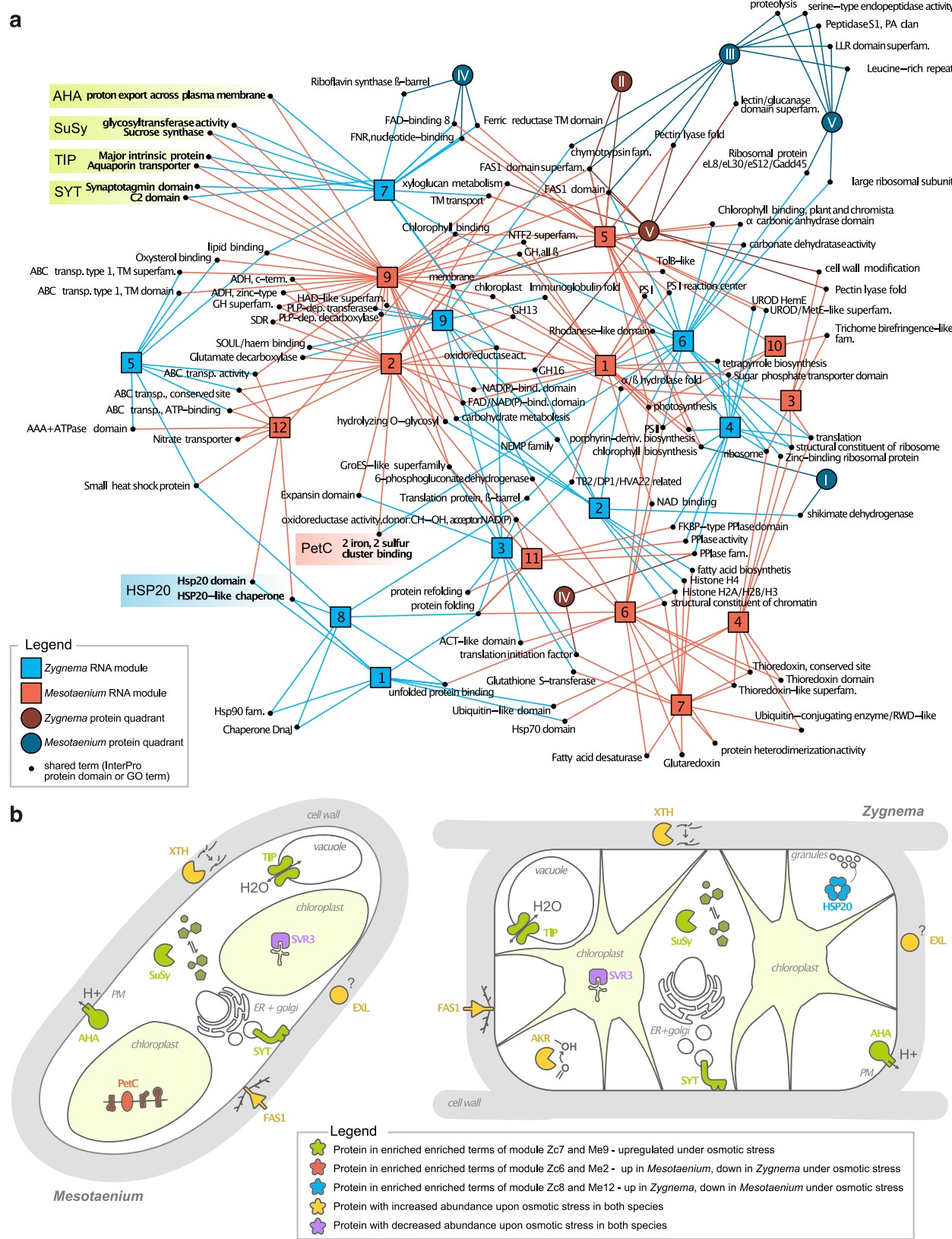

**b**

Legend

- Protein in enriched enriched terms of module Zc7 and Me9 - upregulated under osmotic stress
- Protein in enriched enriched terms of module Zc6 and Me2 - up in *Mesotaenium*, down in *Zygnema* under osmotic stress
- Protein in enriched enriched terms of module Zc8 and Me12 - up in *Zygnema*, down in *Mesotaenium* under osmotic stress
- Protein with increased abundance upon osmotic stress in both species
- Protein with decreased abundance upon osmotic stress in both species

Several terms merit closer examination, particularly two that exhibit opposing regulation under osmotic stress in *Mesotaenium* and *Zygnema*: the Hsp20 domain and 2Fe-2S-binding. These terms encompass multiple proteins, but in both *Mesotaenium* and *Zygnema*, they include Heat Shock Protein Family 20 (Hsp20) and the Rieske iron-sulfur protein PetC, respectively. Hsp20 is a small heat shock protein with a protective role, while PetC, part of the cytochrome b6/f complex, is crucial for electron transport in photosynthesis. Both proteins are involved in stress response mechanisms: Hsp20 prevents the aggregation of proteins and lipids[58], while PetC contributes to photosynthetic redox control[59]. In our experimental setup, *Zygnema* predominantly relies on the

**Fig. 5 | Integrated network of comparative proteomic and transcriptomic analysis in two zygnematophytes. a** This network illustrates the overlap of enriched Gene Ontology (GO) terms and protein domains across transcript modules and/or protein quadrants in *Zygnema* and *Mesotaenium*. Long names have been abbreviated for clarity. Module 7 of *Zygnema* and Module 9 of *Mesotaenium* show the highest overlap in number of shared functional terms, and both exhibiting transcriptional upregulation under osmotic stress. Terms only shared by these two modules are highlighted in green, with representative proteins common to both species written next to the term(s) and schematically depicted in the algae models (see **b**) (AHA = Plasma Membrane H + -ATPase, SuSy = Sucrose Synthase, TIP =

Tonoplast Intrinsic Protein, SYT = Synaptotagmin). Terms and corresponding proteins showing opposing transcriptional regulation under osmotic stress are highlighted in red (upregulated in *Mesotaenium*) and blue (upregulated in *Zygnema*) within the schematic algae model (HSP20 = Small Heat Shock Protein 20, PetC = Component of the Cytochrome b6-f Complex). **b** Schematic Illustration of *Mesotaenium* and *Zygnema* cells with the proteins mentioned above shown. Proteins with significantly increased or decreased abundance (see Fig. 2) in both algae are depicted as well, in yellow and purple, respectively. Locations and functions depicted in the algae model are hypothetical.

Hsp20 family, whereas *Mesotaenium* utilizes PetC, reflecting distinct strategies to cope with osmotic stress.

We attempted to predict Gene Regulatory Networks (GRNs) from temporal transcriptional data using Sliding Window Inference for Network Generation (SWING; supplementary Fig. 3). While the GRNs were not conclusive (likely due to lack of additional and earlier time points), examination of differentially expressed transcripts revealed various signaling components. For instance, the ABA-responsive Element Binding Factor (ABF) was strongly upregulated in both *Zygnema* and *Mesotaenium* (Supplementary Fig. 4).

## Conserved feedback at the cell wall under osmotic and ionic stress

The cell wall is the first barrier of a plant cell. Plants sense the integrity of their cell wall during growth and stress through mechanisms that concertedly maintain cell wall integrity (CWI)[60]. In previous work, we found that homologs that could constitute a CWI sensing and signaling loop co-express in *Mesotaenium*[12,15]. Our combined proteomic and transcriptomic data point to several layers of such biophysical feedback, integrating homeostasis of the cell wall into the overall response (Fig. 5).

Cell wall remodeling is key during acclimation to osmotic challenges[61]. Several cell wall-associated proteins were highlighted in the overlap between both species, including expansins, xyloglucan metabolism and more (Fig. 5). Zygnematophytes are rich in glycosyl hydrolases (GH) akin to land plants[10]. We screened the proteome (FDR < 0.05) and transcriptome (FC > 3) for increases in GHs. While a cellulase (GH5) was identified in the proteome, its increase was relatively modest. The most pronounced increases were observed in the XTHs and enzymes involved in starch degradation, whose transcript were upregulated in the NaCl treatment and the mannitol treatment. To follow up on this, we used LC-MS-based metabolite fingerprinting on *Mesotaenium* and identified a small set of robust features that significantly differed between treatments (Fig. 6a, Supplementary Fig. 5). These features were classified into three main categories: (1) Mannitol, which was expectedly abundant in the mannitol treatment and not detected in the other samples. (2) Sodium formate, a known artifact resulting from the addition of NaCl to biological samples[62]. (3) Carbohydrates build from aldo/keto-hexoses, predominantly detected with sodium adducts in the NaCl treatment. Among the carbohydrate-associated features, several were identified as corresponding to a disaccharide, a trisaccharide, a tetrasaccharide, and a pentasaccharide eluted in the same retention time range. When scanning specifically for glycosides in negative ion mode, several features showed a marked increase in intensity following hyperosmotic treatment. This suggests that *Mesotaenium* may utilize glycosides, such as sucrose or maltose, as osmolytes. Interestingly, the feature corresponding best to proline, often seen as osmolyte in plants[63], did not exhibit any increase upon osmotic treatment, nor were the homologs to the proline biosynthetic proteins P5CS1 and P5CR or the proline transporter BAC2 upregulated.

In the differential abundance analysis, we noted an upregulation of cell wall-situated proteins, including the aforementioned XTHs and cell wall-associated (predicted) proteins containing a fasciclin-like

(FAS1) domain. XTH is a well-characterized enzyme belonging to the GH family 16, which remodels xyloglucan chains. XTH has been frequently reported to be strongly upregulated in response to osmotic stress in land plants[64], assumingly contributing to cell wall strengthening and thereby indirectly preventing water loss[65]. We constructed a phylogenetic framework encompassing all glycosyl hydrolases from GH family 16 (Fig. 6f) and recovered the XTHs (GH16, subfamily 20) exclusively for Embryophyta and Zygnematophyceae—not in any other streptophyte or chlorophyte algae; in Coleochaetophyceae, some proteins were initially annotated as GH16 subfamily 20 using dbCAN3 but were phylogenetically closer to those of subfamily 27. The XTHs from Embryophyta and Zygnematophyceae each form distinct monophyletic groups, in case of the latter further divided into two deep clades (Fig. 6f). Notably, all XTHs responsive to osmotic stress clustered within the same clade.

The FAS1 domain is a conserved structural motif found in extracellular proteins across diverse life forms, with a notable prevalence in plants. This domain provides multiple interaction surfaces, facilitating binding to a variety of ligands. In plants, FAS1 domains occur almost exclusively in the form of fasciclin-like arabinogalactan-proteins (FLA)[66]. Their precise biochemical mechanism is obscure but likely includes interaction and signaling processes within the extracellular matrix[67]. FLAs are a subgroup of the highly *O*-glycosylated plant glycoproteins called arabinogalactan-proteins (AGPs)[68]. They contain one or two (or more) FAS1 domains as well as glycosylated and non-glycosylated intrinsically disordered regions[69]. We examined our significantly differentially abundant proteins for potential AG glycosylation sites and identified one FLA in *Zygnema*, two FLAs in *Mesotaenium*, as well as one pectin methylesterase with potential AG glycosylation sites (Fig. 6c). All can be defined as chimeric AGPs, consisting of a functional, structured protein domain and disordered regions[70]. To investigate the influence of osmotic stress on AGP glycosylation, we analyzed cellular extracts from different treatments using the β-Yariv reagent and various AGP-specific antibodies. In *Mesotaenium*, radial diffusion assays indicated an increase in the galactose backbone of AGPs under salt and mannitol treatments compared to the control (Fig. 6d). Conversely, the two AGP-specific antibodies[71,72] KM1 and JIM13 showed signal decrease in both treatments (Fig. 6d). Both treatments showed significantly elevated levels of hydroxyproline (Hyp) per glycan (Supplementary Fig. 6) corresponding to a more condensed glycan structure with reduced side-chain length while the overall galactose content remains relatively stable. The β-Yariv reagent did not stain *Zygnema* extracts (Fig. 6b), possibly due to differences in AGP glycosylation structure with higher levels of uronic acids (Fig. 6d, Supplementary Fig. 6). Nonetheless, trends in the KM1 antibody results in *Zygnema* were similar to those in *Mesotaenium*. The discrepancy in signals from the backbone and side chains under osmotic stress suggests alterations at the post-translational level. We further scanned upregulated transcripts for glycosyl hydrolases that might hydrolyze AGP side chains. This analysis proved challenging due to the limited functional characterization of many glycosyl hydrolases and the incomplete understanding of AGP glycosylation structure in the investigated algae[73]. Among the identified candidates, GH30—a galactosidase[74]—was upregulated in both species. However, these

proteins lacked signal peptides, complicating their functional interpretation.

We stained live *Mesotaenium* with β-Yariv reagent after 25 h of stress treatment (Fig. 6e, Supplementary Fig. 7). Substantially more staining was observed inside the cells subjected to mannitol treatment. However, when staining with an unrelated dye, fuchsin red, we also noted an increase in intracellular staining, suggesting higher cell membrane permeability in general (Supplementary Fig. 7). This could be attributed to a hypotonic shock following the stress treatment. In contrast, during salt treatment, we frequently observed staining of extracellular material, indicating the release of AGPs into the extracellular space (Fig. 6e). This phenomenon of AGP release has previously been documented in tobacco[75,76]. It was hypothesized that phospholipase C could cleave the glycosylphosphatidylinositol (GPI) anchor of FLAs, thereby releasing them into the extracellular space. However, in our experiments, especially the smaller FLA isoform lacking a GPI anchor was significantly upregulated in protein abundance under salt treatment and simultaneously downregulated at the transcriptional level. Additionally, none of the predicted phospholipase C genes were upregulated, making this mechanism unlikely. To further investigate, we analyzed two modules of transcripts that were strongly upregulated exclusively during salt treatment, Module 3 and Module 10 (Fig. 4), likely reflecting the ionic-specific stress response in *Mesotaenium*. The modules contained some genes related to photosynthesis, many uncharacterized proteins, and pectin methylesterases (PMEs) alongside a PME inhibitor. PMEs are enzymes that demethylate galacturonic acid residues, a key component of the pectin backbone in plant cell walls. Random demethylation loosens the pectin structure, enhancing enzymatic degradation while block-wise demethylation exposes free carboxyl groups on galacturonic acid residues, which can then crosslink with calcium ions to form a gel-like, compact pectin structure[77]. The PME in *Mesotaenium* predicted to have AG sites shares its closest *Arabidopsis* homolog with PME53, and thus likely removes methyl groups in a random manner[78].

Both pectin and AG groups exhibit a strong affinity for binding calcium ions. However, sodium ions, introduced in large quantities during NaCl treatment, can compete with calcium for these binding sites[79]. This competition can disrupt the calcium-mediated crosslinks between pectin molecules, potentially resulting in cell wall softening and destabilization[80]. During long-term ( > 26 days) NaCl treatment, we observed significant deterioration of the cell walls in *Zygnema* (Fig. 6b). While the algae survived, no growth was recorded under these conditions. In contrast, *Mesotaenium* showed remarkable resilience: it remained healthy, maintained active growth, and preserved chloroplasts with a proper shape and vibrant green pigmentation. Notably, some differences were observed compared to the control treatment, including the formation of small filaments up to eight cells in length and an increased accumulation of cell wall remnants (see Fig. 6b and Supplementary Fig. 8).

Overall, our data pinpoint deeply conserved proteins that zygnematophyte cells use in homeostatic mechanisms—that include cell wall remodeling—to successfully acclimate to osmotic stress (Fig. 7).

## Discussion

The evolutionary origin of land plants is a complex question that has been addressed from multiple angles[81–84]. One angle is to infer cell biological features that land plants share with zygnematophytes, and that might have aided the earliest land plants in their conquest of land. A glaring feature is the shared capacity of the streptophyte cell to withstand terrestrial challenges[12,85]. Water scarcity that leads to osmotic stress is one of the most palpable challenges in the terrestrial habitat and a subject of several physiological studies in zygnematophytes[86–90].

We pinpoint systems biological responses to osmotic stress in the zygnematophyte *Mesotaenium* and *Zygnema* (Fig. 7). Overall, the responses differed markedly between the two algae, yet led to acclimation to hyperosmotic challenges in both systems. Further, in both cases, the responses were built on a deeply conserved protein chassis. In *Mesotaenium*, the control treatment—serving as a mock treatment conducted alongside the tested treatments for comparison—exhibited minimal changes over time. In contrast, *Zygnema* showed a pronounced and temporally dynamic response under the control conditions over the course of 25 h. This highlights the resilient internal and diurnal growth program of *Zygnema*. It seems that *Zygnema* prioritizes growth and rapid adaptation to improved conditions rather than immediate responses to worsening stress. This is consistent with other observations highlighting the resilience of *Zygnema*[14,40,90]. Another striking difference between the species was their response to salt and mannitol treatments. In *Zygnema*, the response to salt generally fell between the control and mannitol treatments in magnitude, consistent with the varying degrees of osmotic stress. By contrast, in *Mesotaenium*, the salt-induced response was often comparable to, but faster than, the response to mannitol. Moreover, *Mesotaenium* exhibited a distinct reaction exclusive to the salt treatment, suggesting that ionic components not only accelerate the hyperosmotic stress response but also trigger unique ionic-specific pathways.

In *Mesotaenium*, many genes specifically upregulated under salt stress remain unidentified, one notable exception being PME, which modifies and thereby regulates pectin structure in the cell wall. Another significant observation is the release of AGPs into the medium under salt stress, a phenomenon also reported previously in tobacco[75,76]. A physicochemical property of both pectin and AGPs is their potential to bind to positively charged ions. Pectin can form complexes with calcium to crosslink its own polymers[91] and AGPs, particularly their arabinogalactan (AG) domains, are known to function as calcium reservoirs due to their strong calcium-binding properties[79,92]. Under NaCl treatment, sodium ions could compete with calcium ions for binding sites in both calcium-pectin complexes and calcium-AG interactions (Fig. 6g). This competition likely drives an increase in block-wise PME activity to stabilize the cell wall structure. Furthermore, the AG regions of *Mesotaenium*'s PME53 may serve as ion sensors or calcium reservoirs. The increased tendency of *Mesotaenium* to form short filaments under prolonged salt treatment remains unclear but may also be related to alterations in pectin-modifying enzymes, as pectin plays a crucial role in cell adhesion and the formation of multicellular complexes[77].

AGPs, particularly the FLAs, are influenced not only by ionic stress but also by other types of stress like osmotic stress. Just recently, some FLAs from *Physcomitrium patens* were shown to be involved in the cellular response to rapid dehydration stress[93]. FLAs might play a structural role in the cell wall, and alterations in glycosylation patterns could potentially modify cell wall properties[94]. Alternatively, FLAs have been proposed to play significant roles in signaling[66] or to be transported via vesicles to the vacuole under osmotic stress to foster ion transport[76]. If this mechanism occurs, FLAs might be degraded within the vacuole, thereby increasing the osmotic potential inside it and helping to restore the cell's turgor pressure. While this is an intriguing hypothesis, it warrants further detailed research to elucidate the underlying mechanisms.

The vacuole is key in osmotic stress responses. Tonoplast intrinsic proteins (TIPs), which facilitate the transport of water and small solutes, are upregulated under such conditions. Mannitol treatment appears to visibly alter the tonoplast vacuolar system in *Mesotaenium*. Similar changes, characterized by a transition from a simple globular vacuolar shape to a more network-like structure, have been reported in tobacco under osmotic stress[95]. In *Zygnema*, this shift in vacuolar structure was less evident. However, changes in chloroplast morphology were observed, indirectly suggesting vacuolar shape alterations. The star-shaped chloroplasts, seemingly anchored to the cell wall, lose their distinctive shape leading up to and during plasmolysis.

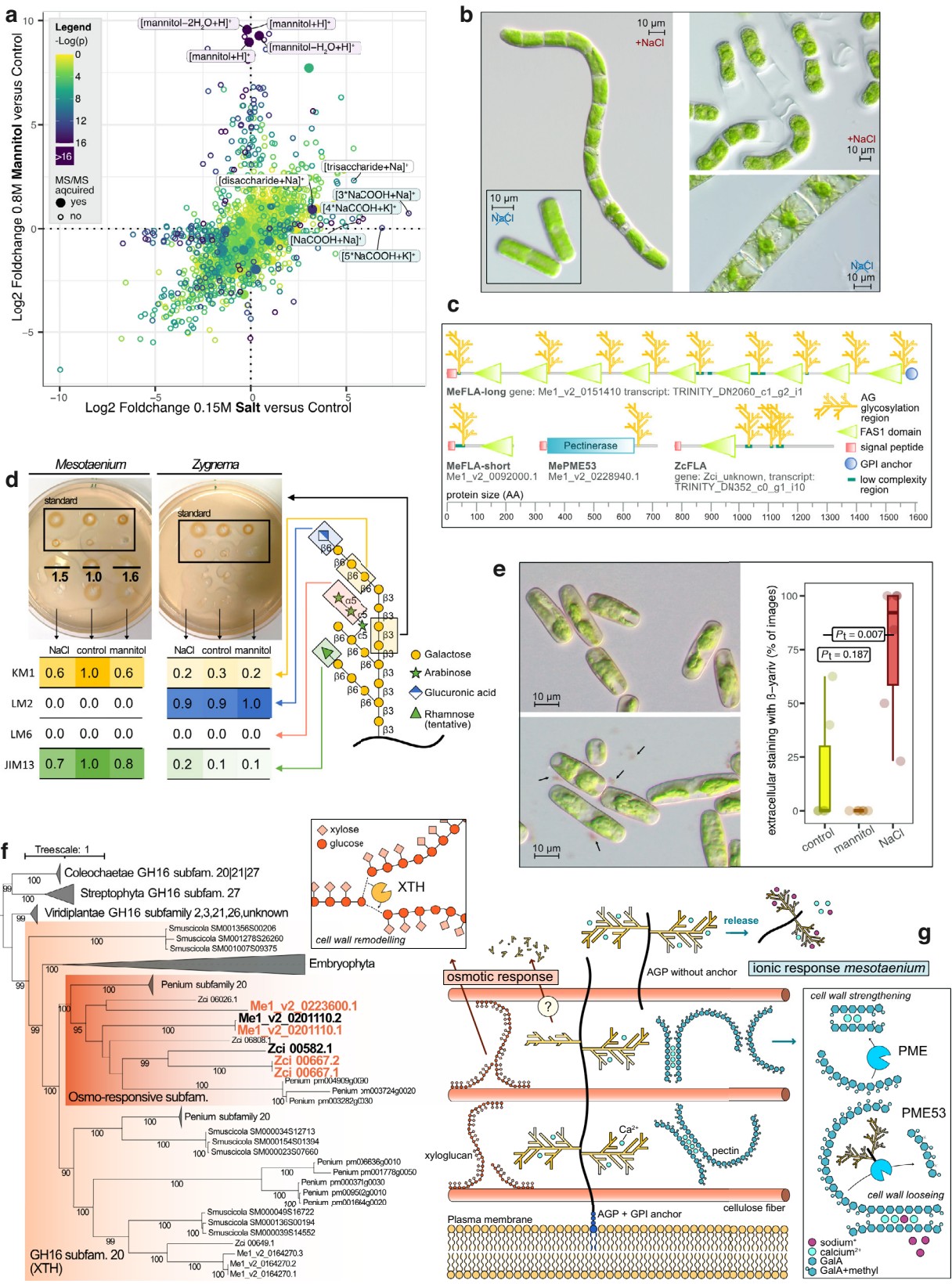

In the zygnematophyte *Penium*, similar vacuolar shape changes were observed under extreme hyperosmotic stress[43]. Such alterations in the vacuolar system have been hypothesized to signal the onset of programmed cell death or to serve as a means of membrane storage until conditions become more favorable[43,95]. We propose that these vacuolar changes may also play a role in modifying cellular structures to either prevent or prepare for plasmolysis.

We identified numerous proteins that are strongly upregulated under osmotic stress, including several well-established abiotic-stress-responsive genes in plants, such as AKR, AHA, LEA, XTH, and

**Fig. 6 | Effect of osmotic and ionic stress on sugar and cell wall biology of zygnematophytes. a** Log2 fold changes of metabolites from metabolite finger-printing analysis of methanol-extracted samples after 9 hours of stress treatment with 0.8 M mannitol versus 0.15 M NaCl in positive ion mode. Identified metabolites are labeled. Log(*P*-values) are based on multiple sample test ANOVA with a permutation-based FDR set to 0.05 and S0 set to 0.15. **b** Long-term (26 days) effects of 0.15 M NaCl treatment on algae morphology. Left: *Mesotaenium*. Right, top: *Zygnema* after prolonged treatment. Right, bottom: *Zygnema* morphology after recovery for 14 days after 14 months of 0.15 M NaCl treatment. Pictures are representatives of six independent biological replicates of time-course experi-mentation; see supplemental Figure 8 for details. **c** Domain architecture, names, and IDs of proteins significantly more abundant (FDR < 0.05) during 0.15 M NaCl and/or 0.8 M mannitol treatments that have predicted arabinogalactan glycosyla-tion sites. Species abbreviations: Me = *Mesotaenium*, Zc = *Zygnema*. **d** Arabinogalactan protein (AGP)-focused analysis of aqueous-extracted samples after 25 hours of 0.15 M NaCl or 0.8 M mannitol treatment. Top: β-Yariv radial dif-fusion assay. Bottom: Sugar epitope ELISA results. Signal intensities are normalized individually for each antibody to its respective maximum measured value. Right:

Schematic representation of an AGP chain (not representative for *Mesotaenium* and *Zygnema*). **e** Staining of *Mesotaenium* with β-Yariv reagent after 25 hours of 0.15 M NaCl treatment, with quantification of extracellular β-Yariv-reactive material across different treatments plotted as box plot, where the box shows the interquartile range (IQR) from the 25th (Q1) to the 75th percentile (Q3) while the whiskers extend to minimum and maximum values in the data determined by Q1 − 1.5 × IQR to Q3 + 1.5 × IQR (data points outside are defined as outliers) the center line repre-sents the mean. Each data point (see jitter) represents a biological replicate ($n = 6$). Source data are provided as a Source Data file. **f** Phylogenetic tree of glycosyl hydrolase family 16. The expanded clade represents subfamily 20, which includes xyloglucan endotransglucosylase/hydrolases (XTHs). The gene IDs in bold black indicate transcripts upregulated (ΔFC > 3) during osmotic stress, while those highlighted in orange correspond to proteins that are significantly more abundant (FDR < 0.05) under osmotic stress conditions. **g** Schematic representation of hypothesized major changes in cell wall components during osmotic and ionic stress in *Zygnema* and *Mesotaenium*. The respective enzymes showing significant changes at both protein and transcript levels are indicated. The depicted cell wall structure is hypothetical.

ELIP[37,65,90,96,97]. Additionally, SuSy and enzymes from the starch degra-dation pathway were upregulated upon osmotic stress, potentially playing a role in osmolyte production. Consistent with findings in other streptophyte algae, previous studies have shown that sucrose synthase (SuSy) activity is significantly elevated in *Klebsormidium cre-nulatum* under desiccation stress[98]. Similarly, sucrose levels were found to be markedly increased in *Chara braunii* in response to salinity-induced stress[99]. Investigating species-specific patterns revealed more sugar-degrading enzymes, such as pullanase in *Meso-taenium*. Our data point to various biophysical feedback mechanisms enriched in the hyperosmotic stress response, including SYTs, which have only infrequently been associated with osmotic stress[100]. A fur-ther link to the cell wall and growth exists through EXL homologs, which were consistently enriched and facilitate growth under chal-lenging conditions[101,102]; indeed, EXL consistently appears as a stress hub shared between land plants and zygnematophytes[14,15]. This goes hand in hand with the changes in OSCA, one of the key mechan-osensitive ion channels[103], and a major hub in a predicted gene reg-ulatory network shared by zygnematophyceae and land plants[14], as well as the cell wall-loosening protein expansin[104].

Overall, our comparative approach describes multiple layers of biophysical feedback that concertedly acclimate cell wall and cell interior through mechanisms shared by land plants and their algal sisters when facing a primary terrestrial stressor. Our results point to a core hyperosmotic stress response chassis present and functional in the last common ancestor of Zygnematophyceae and land plants, predating the emergence of the latter. Among the Zygnematophyceae responses, we observed several proteins with no known function that were nonetheless strongly induced at the transcript or protein level, pointing to previously unrecognized components of algal stress tol-erance. Land plants, in turn, possess their own lineage-specific adap-tations, such as changes in root architecture and stomatal regulation[28], which are naturally absent in algae lacking three-dimensional growth. Focusing on the cellular level alone, the identified innovations under-lying hyperosmotic stress tolerance appear to have originated before the rise of the embryophytes (Fig. 7d), suggesting that their last common ancestor was frequently exposed to this terrestrial stressor and thus underscoring that there were likely multiple transitions, back and forth, between terrestrial and aquatic habitats throughout strep-tophyte evolution[105–109].

## Methods
### Cultivation of the algae
*Mesotaenium endlicherianum* SAG 12.97 and *Zygnema circumcar-inatum* SAG 698-1b were sourced from the Culture Collection of Algae at Göttingen University (SAG). They were cultivated on cellophane

disks[90], with the modification that *Mesotaenium* was grown on 1.5% agar-solidified C medium[110] for a total of 8 days at 60-80 µmol photons m$^{-2}$ s$^{-1}$ and *Zygnema* was grown on 1% agar-solidified WHM medium (MZCH recipe) for a total of 14 days at 15-20 µmol photons m$^{-2}$ s$^{-1}$.

### Stress treatments
The stress treatments were performed under the same conditions as the algae cultivation. After two hours of light exposure within their 16/8 h light-dark cycle, the algae cultures were transferred to a sterile workbench. Within a maximum of 15 minutes—during which minor fluctuations in light, temperature, and humidity could occur—the cel-lophane discs with algae were placed onto fresh plates containing the same medium used for cultivation, supplemented with either no additive (control), 0.15 M NaCl, or 0.8 M D-mannitol. For the C-medium and WHM medium, respectively, this resulted in osmolarities of 0.015 Osm/L and 0.018 Osm/L for the control medium, 0.315 Osm/L and 0.318 Osm/L for the salt medium, and 0.815 Osm/L and 0.818 Osm/L for the mannitol medium, with the contribution of agar (which consists of long polymers) to the total osmolarity neglected. The treated algae were then returned to the cultivation chamber to resume growth.

### Physiological measurements
At 0, 3, 6, 9, and 25 hours during the stress treatments, algae samples were incubated in the dark for $20 \pm 5$ minutes. Each sample was then placed on the sensor of the Walz PAM-210 Chlorophyll Fluorometer, and the maximum quantum yield of PSII (Fv/Fm) was measured. The following instrument settings were used: ML = 6, AL = 3, FR = 8, and SP = 10. For each biological replicate, two technical replicates were measured, and the mean Fv/Fm value was recorded. Algal samples used for PAM measurements were excluded from further stress experiments to prevent potential measurement interference.

### Microscopy
At 25 hours, algae samples were transferred to a liquid medium, and microscopic images were immediately captured using a Zeiss Axio-scope 7 light microscope at 400× magnification. For *Mesotaenium*, individual cells were counted, while for *Zygnema*, individual filaments were enumerated. Additionally, the presence or absence of specific traits was recorded. For each trait, the percentage of cells or filaments exhibiting the trait was calculated per biological replicate. Detailed definitions of all traits, along with the number of cells or filaments analyzed, are provided in the Source Data.

For microscopic analysis using the β-Yariv reagent, treated algae (control, 0.15 M NaCl, or 0.8 M D-mannitol) were returned to the growth chamber and incubated for 25 hours. Staining was performed with β-Yariv reagent and 0.05−1% Fuchsin stain, the latter serving as a

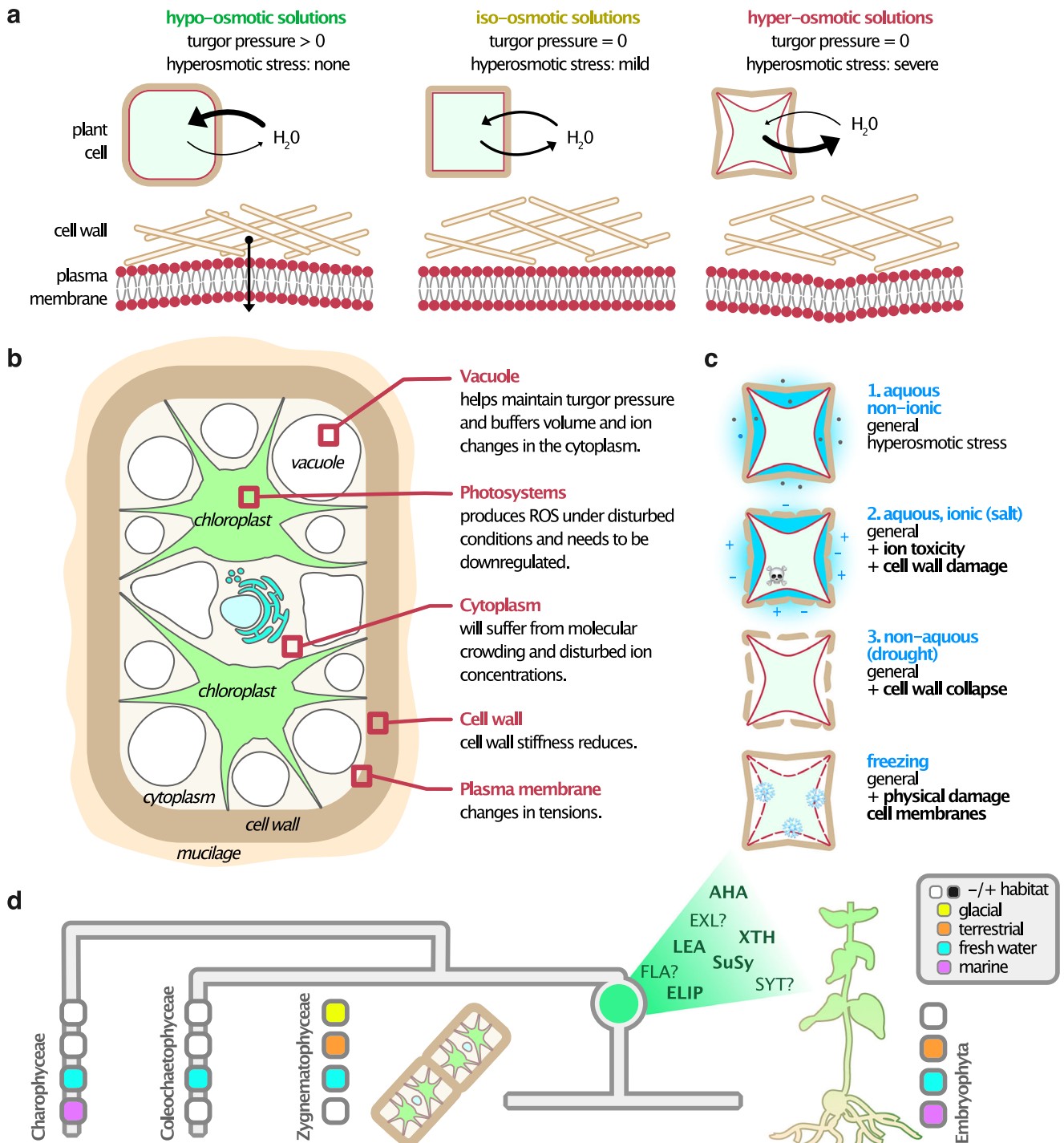

**Fig. 7 | Schematic overview of plant cell responses to hyperosmotic stress.**
**a** Illustration of a plant cell under varying osmotic conditions. Plant cells typically function optimally in mildly hypo-osmotic environments, which support the build-up of turgor pressure. Under iso-osmotic or hyperosmotic conditions, turgor is lost, resulting in decreased plasma membrane tension, reduced cell wall stiffness, and reduced cell volume. **b** Detailed view of a plant cell (specifically a zygnematophycean alga) highlighting key subcellular compartments and cellular structures involved in the response to osmotic stress. **c** Types of hyperosmotic stress: (1) non-ionic osmolarity increase of the environment in the presence of water, (2) increased ionic solute concentration, (3) water loss due to drought, or (4) freezing, which reduces water availability. **d** Cladogram of the Phragmoplastophyta highlighting key players in hyperosmotic stress responses in Zygnematophyceae that are also present in embryophytes. Proteins with established roles in embryophyte hyperosmotic stress are shown in bold, and those requiring further study are marked with a question mark.

control for nonspecific staining. Algae were then incubated with 100–200 µM β-Yariv dissolved in C-medium for 10 minutes. Imaging was conducted as described above. To ensure direct comparison, images of each biological replicate were acquired in a staggered manner over approximately 10 minutes. For quantification, additional imaging was carried out for up to 2 hours, with treatments observed for about 3 minutes each. A total of 176 photos were captured, and extracellular staining was quantified by assessing the presence or absence of red-brown staining of extracellular material in each image (see Source Data).

## Measurements of AGPs, sugar, and weight

After 25 hours of stress treatment, eight technical replicates (equals eight petri dishes) were collected into Eppendorf tubes (making one biological bulk replicate) and immediately frozen in liquid nitrogen. The samples were then lyophilized using a VaCo 2 Zirbus lyophilizer at −80 °C. The weight of each Eppendorf tube was recorded at three stages: empty, with the frozen sample, and with the lyophilized sample. Two biological bulk replicates were used for analyses. The lyophilized samples were ground with a beat-mill (VWR Star-Beater, VWR International, UK) and pre-extracted two times (2 h and 21 h, respectively) with acetone:water (70:30, v/v) in a ratio of 1:100 (v/v) at 4 °C on a shaker. Afterwards, the samples were extracted with double-distilled water for 21 h at 4 °C on a shaker. In between all extraction steps, the supernatant was separated from the pellet by centrifugation at 19,000 × $g$, 20 min, 4 °C (Heraeus Multifuge X3R, Thermo Fisher Scientific Corp., USA). The resulting aqueous extract was lyophilized (Christ Alpha 1-4 LSC, Martin Christ GmbH, Germany) and afterwards dissolved at a concentration of 10 mg/ml under homogenization in the beat-mill (VWR Star-Beater, VWR International, UK, 10 Hz, 1 h with a metal beat). All following experiments were performed with aliquots of this solution.

Total sugar and uronic acid contents were determined colorimetrically by the protocols of Dubois et al.[111] and Blumenkrantz and Asboe-Hansen[112], respectively. Radial gel diffusion assay for interaction of samples with β-D-glucosyl-Yariv (β-D-GlcY) reagent was performed by using the protocol of Castilleux et al.[113] with modifications. In brief, 20 µl of the aqueous extracts were loaded into cavities of an agarose gel containing 0.25 mol/l NaCl and 10 µg/ml β-D-GlcY. Gum arabic solutions in different concentrations (1.0, 0.5, 0.25, 0.125, 0.063 mg/ml in double-distilled water) were used as controls. Hydroxyproline content was determined by colorimetric quantification using the protocol of Stegemann and Stalder[114]. Reactivity of aqueous extracts with the monoclonal antibodies KM1, LM2, LM6, and JIM13 was determined by enzyme-linked immunosorbent assays following the workflow of Mueller et al.[115]. Neutral monosaccharide composition was determined the derivatization of Blakeney et al.[116] with modifications (see Mueller et al.[115]). To determine mannitol content, the hydrolysis and reduction steps of the derivatization were omitted. All experiments were performed in three technical replicates per biological bulk replicate.

## Transcriptomics

At 0, 3, 6, 9, and 25 hours, algae samples were collected and immediately frozen in liquid nitrogen, followed by storage at −70 °C. For each time point and biological replicate (5 replicates in total), two halves of a culture plate were sampled using a spatula, pooled into an Eppendorf tube, and frozen in liquid nitrogen. The remaining plate halves were reserved for proteomics or metabolomics analyses. RNA was extracted using the Spectrum™ Plant Total RNA Kit following the supplemented protocol A, with modifications made to the cell lysis step: Zygnema samples were lyophilized for 48 hours, then ground using a Retch mill at 20 Hz for 1 minute. One milliliter of lysis buffer was added to the ground, dried samples, which were vortexed for 40 seconds and heated at 56 °C for 5 minutes. Mesotaenium samples were placed in 500 µL of lysis buffer, vortexed for 30 seconds, sonicated in a DK SONIC™ ultrasonic cleaner, and heated at 56 °C for 5 minutes. Following lysis, DNase treatment with DNase I was performed. For Zygnema, this step was carried out directly on the binding column, while for Mesotaenium, it was performed post-elution of the binding column.

RNA sequencing, as well as quality control and raw data trimming, were conducted according to the methods detailed in Dadras et al.[15]. The sample "salt_9hours_BR2" from Zygnema was excluded from further analysis due to a low read count (<0.1 million reads). Trimmed reads were mapped to the transcriptome (see section *Transcript-Based Proteome Assembly* for details) using Kallisto with bootstrap-samples

set to 100. Quantified reads were adjusted in Sleuth to account for technical variability in transcript quantification, and the transcripts per million (TPM) values were subsequently exported. Within the Sleuth visualization environment, the distribution of TPM values was examined. Based on this analysis, the sample "mannitol_3hours_BR3" from Mesotaenium was excluded from further analysis due to an abnormal TPM distribution. Transcripts were filtered based on abundance thresholds to ensure data quality. Transcripts where a single sample had a TPM value below $2^{-6}$ and the average abundance across all samples was less than 2 TPM were removed. Additionally, transcripts with more than five values below $2^{-6}$ and an average abundance across all samples below 4 TPM were excluded from the dataset. Remaining TPM values below $2^{-6}$ were set to $2^{-6}$. All TPM values were log-transformed prior to further analysis to normalize the data.

## Proteomics

Proteomics analysis was performed on the transcriptomics-paired samples from Mesotaenium (0, 6, and 25 hours) and Zygnema (6 and 25 hours). Protein extraction followed an adapted version of the methods of Niu et al.[117], with species-specific protocols. For extraction, each sample was homogenized in extraction buffer (Mesotaenium buffer: 1 mL/sample, 50 mM Tris-HCl, pH 7.5, 10 mM KCl, 0.2 mM PMSF; Zygnema buffer: 1.5 mL/sample, 1% SDS, 0.1 M Tris-HCl, pH 6.8, 2 mM EDTA-Na$_2$, 20 mM DTT, 2 mM PMSF). Samples were ground in a Ten Broek grinder on ice for 1 minute (Mesotaenium) or 4 minutes (Zygnema), transferred to clean 1.5 mL eppendorf tubes, and centrifuged (Mesotaenium: 1000 × $g$ for 10 seconds; Zygnema: 10,400 × $g$ at 4 °C for 7 minutes). The supernatant was then processed further. For precipitation, 100 µL (Mesotaenium) or 2 × 200 µL (Zygnema) of supernatant was added to 1 mL of pre-cooled ethanol (Mesotaenium) or acetone with 5 mM DTT (Zygnema), briefly vortexed, and incubated overnight at 4 °C (Mesotaenium) or −20 °C (Zygnema). After precipitation, samples were centrifuged (Mesotaenium: 2000 × $g$ at 4 °C for 15 minutes; Zygnema: 5000 × $g$ at 4 °C for 10 minutes) and the supernatant discarded. The pellets were washed twice with pre-cooled ethanol (Mesotaenium) or acetone with 5 mM DTT (Zygnema). Mesotaenium samples were dried in a speed-vac at 50 °C for 30 minutes, while Zygnema samples were air-dried for 5−8 minutes. The pellets were dissolved in either 30 µL of 5% SDS with 6 M urea (Mesotaenium) or 100 µL of the extraction buffer (Zygnema). Protein concentration was determined using a 20× dilution of the protein solution in water and the Thermo Scientific™ Pierce™ BCA Protein Assay Kit. We noticed that the Zygnema protocol, particularly the acetone-based precipitation, also performed slightly better for Mesotaenium compared to ethanol precipitation.

Protein samples (50 µg) were loaded onto 12% polyacrylamide SDS-PAGE gels (1 mm thickness) and electrophoresed at 100 V using a Bio-Rad™ Mini-PROTEAN® 3 Cell system. The gels were run until the dye front had migrated approximately 1.8 mm into the separation gel. Following electrophoresis, gels were briefly rinsed with distilled water and incubated for 60 minutes in a fixation solution containing 40% (v/v) ethanol and 10% (v/v) acetic acid. For staining, gels were immersed in Coomassie Brilliant Blue stain (0.1% (w/v) Coomassie Brilliant Blue G-250, 5% (w/v) aluminum sulfate-(14−18)-hydrate, 10% (v/v) ethanol, and 2% (v/v) ortho-phosphoric acid) for 30 minutes. Excess stain was removed by washing the gels three times with distilled water for 10 minutes per wash. All staining and washing steps were performed on a rocking platform. Protein-containing gel pieces were excised using a scalpel and subjected to an overnight wash in fixation solution, followed by three additional washes with distilled water (10 minutes each) on a rotating platform. Proteins were subsequently digested and peptides extracted in-gel according to the method described by Shevchenko et al.[118]. Peptides were then desalted, and LC-MS/MS analysis was performed as described in the methods of Niemeyer et al.[119].

For initial quality control of the acquired LC-MS/MS data, the raw files were converted to mzML format using MSconvert with the Peak-Picking algorithm. Two-dimensional LC-MS chromatograms were plotted in MSnbase to identify abnormalities in the spectra. Based on this assessment, the sample "mannitol_25hours_BR3" from *Zygnema* and the samples "mannitol_25hours_BR1" and "0hours_BR2" from *Mesotaenium* were excluded from further analysis. Subsequently, the raw files were analyzed using MaxQuant with mono on a high-performance computing cluster. Quality reports were generated with the R package PTXQC in two consecutive runs. The first MaxQuant analysis aimed to identify batch effects and discover novel protein groups (details provided in the "Transcript-Based Proteome Assembly" section and a schematic overview of the data-analysis in Supplementary Fig. 9). During this analysis, it was observed that the variation of *Mesotaenium* samples in the PCA was more strongly influenced by batch effects than by the experimental treatment. As a result, samples from *Mesotaenium* originating from two SDS-PAGE gels were excluded from the second run. For the second MaxQuant run, the updated FASTA database was employed, and LFQ as well as iBAQ values were calculated. The LFQ values were imported into Perseus for further processing. These values were divided into three distinct matrices: 1) A matrix for the 6-hour timepoint (used for 6-hour differential expression analysis). 2) A matrix for the 25-hour timepoint (used for 25-hour differential expression analysis). 3) A comprehensive matrix containing all samples (used for PCA analysis). Within each matrix, protein groups in which all conditions contained ≥1 missing value were excluded. For the remaining missing values, imputation was performed using random sampling from a normal distribution (downshift = 1.9, width = 0.3). The resulting imputed data were utilized for differential abundance analysis, relative protein-RNA comparison, and functional enrichment analysis. The iBAQ values, in contrast, were not processed in Perseus. Instead, they were directly used for absolute protein-RNA comparisons.

## Metabolomics

The transcriptomics-paired samples from *Mesotaenium* (3 and 9 hours) were lyophilized using a VaCo 2 Zirbus lyophilizer at −80 °C. The dry mass of each sample was adjusted to 9–11 mg per technical replicate. Two types of metabolite extracts were prepared for subsequent LC-MS analysis: methanol-based extracts and lipid extracts. For the methanol-based extracts, 1 mL of 70% (v/v) methanol (pre-cooled to 8 °C) was added to the lyophilized, ground samples. The samples were ultrasonicated twice, each for 15 minutes. Following ultrasonication, the samples were centrifuged at $16,000 \times g$ for 20 minutes at 4 °C. From the resulting supernatant, 200 μL was transferred to a vial containing an insert. The extracts were dried using a SpeedVac concentrator and reconstituted sequentially: first in 20 μL of 100% methanol, followed by 40 μL of $H_2O$, with vortexing after each addition. The reconstituted extracts were stored at 4 °C until LC-MS analysis. For the lipid extractions, 600 μL of tert-butyl methyl ether (MTBE):methanol (2:1, v/v) was added per 10 mg of lyophilized, ground sample. The mixture was vortexed and then sonicated for 15 minutes. After adding 100 μL of $H_2O$, the samples were vortexed again and centrifuged at $14,000 \times g$ for 20 minutes at 4 °C. From the upper (MTBE) phase, 200 μL was transferred to a vial containing an insert and dried using a nitrogen evaporator. The dried extracts were reconstituted by adding 40 μL of methanol, followed by 10 μL of $H_2O$, with vortexing after each addition. Extracts were analyzed by LC QTOF MS as described in Kasper et al.[120].

The raw Agilent ".d" data files were converted to ".mzML" format using MSConvert with the PeakPicking algorithm applied. The resulting ".mzML" files were processed in MZmine 4 through the following sequential steps: (1) Mass Detection: Initial detection of mass-to-charge (m/z) values, (2) Chromatogram Builder: Construction of chromatograms based on m/z values and retention time. (3) Local Minimum Feature Resolver: Refinement of chromatographic peaks to resolve overlapping features. (4) Join Aligner: Alignment of features across samples to account for retention time shifts. (5) Feature Filtering: Retention of features based on minimal intensity thresholds and presence in a minimum number of samples. (6) Gap Filling: Use of the Peak Finder algorithm to fill gaps in feature matrices where peaks were not initially detected. The identified features were exported and imported into Perseus for statistical analysis. Within Perseus, the data were log-transformed, missing values were imputed from a normal distribution (width = 0.3, downshift = 1.8) or with the constant value of 1000 when combining features from both ion modes, and an ANOVA test with FDR-correction was conducted to identify statistically significant differences between groups (see Supplementary Data 4). Additionally, autoMSMS files where available, were integrated into the MZmine analysis to match fragmentation spectra with their corresponding features. However, their quantification data were excluded from the statistical analysis in Perseus.

## Transcript-based proteome assembly

A transcriptome for both *Zygnema* and *Mesotaenium* was assembled using Trinity, incorporating the samples "mannitol_9hours_BR1," "salt_3hours_BR4," and "salt_25hours_BR5." The quality-trimming parameters, adapter sequences, and settings applied during trimming were consistent with those used in Trimmomatic during the regular transcriptomics analysis. Subsequently, Kallisto was employed to quantify the read counts for the newly assembled transcriptome across the aforementioned samples. Transcripts with read counts below 2 TPM for *Mesotaenium* and below 3 TPM for *Zygnema* were excluded from the transcriptome (see Supplementary Data 5 for assembled transcriptomes). Coding sequences were then identified using TransDecoder with default settings. For *Mesotaenium*, the --complete_orfs_only flag was applied, ensuring only complete open reading frames were included. This setting was unintentionally omitted for *Zygnema*, leading to the grouping of incomplete translations with their corresponding complete transcripts for gene ontology (GO) enrichment analysis. The inferred proteome was analyzed using MaxQuant, incorporating.raw files from test runs and samples disqualified due to batch effects. This analysis was performed with the following existing protein databases: *Mesotaenium* assembly version 1 with annotation version 2 gene models[15] and *Mesotaenium* chloroplast-encoded proteins[121] for *Mesotaenium*, and *Zygnema circumcarinatum* SAG698-1b (the exact strain used for experimentation) gene model[10] and UTEX1559 (a strain relative) plastome[122] for *Zygnema*. Only protein groups with at least one unique peptide and a Q-value of 0 were retained. Protein groups exclusively originating from the newly transcriptome-based proteome were incorporated into the standard proteome to generate an updated database. This database was subsequently assessed for completeness and quality using BUSCO with the -l viridiplantae_odb10 dataset.

## Downstream analysis of protein and transcript data: differential abundance analysis

The log2 fold changes of the processed LFQ values were calculated for mannitol-treated samples versus control samples and NaCl-treated samples versus control samples. These calculations were performed for both the 6-hour and 25-hour time points. The data were visualized using the ggplot package in R, with points color-coded by p-values. P-values were computed per matrix in Perseus using an ANOVA test with parameters S0 = 0.15 and permutation-based FDR = 0.05 to determine significance. Significant protein groups were manually annotated using blastp (default settings, databases nr and SwissProt), domain annotations from SMART, and, where necessary, results from TargetP, dbCAN3, and NetGPI. For the filtered TPM values, log2 fold changes were computed for each combination of NaCl-treated and mannitol-treated time points relative to their respective controls. Statistical analysis was conducted using Python's scipy.stats and statsmodels

packages, where one-way ANOVA tests were performed, followed by FDR correction using a multitest adjustment. These data were visualized similarly to the LFQ results, but with the Seaborn package in Python.

### Downstream analysis of protein and transcript data: comparative analysis

Transcript and corresponding protein levels were compared using both relative and absolute approaches. For the relative comparison, the computed log2 fold changes for LFQ and TPM values from the differential abundance analysis were directly plotted against each other without further processing. Additionally, log2 fold changes of the integrated TPM levels for all transcripts were calculated for each treatment-control pair and plotted against the log2 fold changes for LFQ values as well.

For the absolute comparison, transcript and protein group levels were compared on a sample-by-sample basis. This was feasible as the RNA sequencing and shotgun proteomics were performed on the same biological samples. Biological replicates were analyzed individually without grouping. Protein levels were quantified using iBAQ values, which were ranked within each sample, with a rank of 1 assigned to the most abundant protein group. The TPM levels were ranked in the same way. The ranks of the protein groups were then matched with the ranks of their corresponding transcripts, and the relationships were visualized using the R package ggplot2. In cases where a protein group corresponded to multiple transcripts, the transcript with the highest abundance was selected for comparison. For the *Mesotaenium* dataset, each protein-transcript pair was further annotated by coloring points according to contig length from the genome[8].

### Downstream analysis of protein and transcript data: functional enrichment

InterProScan[123] results published in Rieseberg et al.[14] were utilized for Gene Ontology (GO) term and InterPro domain annotation. Newly identified transcripts were also annotated using InterProScan. For the functional enrichment analysis at the protein level, the data from the 6-hour and 25-hour time points were merged. Proteins were grouped into nine quadrants based on their log2 fold change (log2FC), using thresholds of 1 and −1. These protein groups were then mapped back to the gene level, and functional enrichment was performed for all quadrants except the central one (where $|log2FC| < 1$). The analysis was conducted using FuncE with an e-value cutoff of <0.01.

To analyze transcript functional enrichment, a time-dependent clustering approach was employed. Log2 fold changes were computed for each treatment-timepoint combination relative to timepoint 0. Transcripts were filtered based on criteria: max(tpm) > 5, max(log2 fold change) > 1, and adjusted p-value < 0.01. The filtered log2 fold changes were clustered using a Gaussian Mixture Model (GMM) implemented in Python's Scikit-learn library (sklearn.mixture.GaussianMixture). To determine the optimal number of clusters, the Bayes Information criterion (BIC) were calculated for models with 1 to 100 clusters, and the configuration with the lowest BIC score was selected. Transcripts were then grouped into temporal patterns based on the selected number of clusters (see Supplementary Fig. 10). Next, these patterns were normalized, and the clustering process was refined with GMM. Starting with 15 clusters, the number was progressively reduced until two seemingly distinct patterns were merged, indicating the final grouping. These final transcript groups, referred to as modules, were mapped back to their corresponding genes. Functional enrichment analysis of these gene modules was performed using FUNC-E with an e-value cutoff of <0.01.

An overview was created of enriched terms showing overlap between transcript modules and/or protein quadrants from *Zygnema* and/or *Mesotaenium* (see Zenodo). Terms associated with only a single gene were excluded, as were terms with 100% gene overlap. In cases of identical overlap, the term with the best p-value was retained; if p-values were equal, one term was selected at random. The resulting list of terms was visualized as a network using the igraph package in R.

### Phylogenetic analysis XTH

The protein data for 16 embryophytes, 11 streptophyte algae, and 8 chlorophytes, building on the dataset used in de Vries et al.[124], were combined into a single database, incorporating the updated *Mesotaenium* gene model[15]. CAZyme families were identified using dbCAN3 with default settings. Based on results from the HMMer search, proteins identified and annotated as Glycosyl Hydrolase family 16 (GH16) or any of its subfamilies were selected for phylogenetic analysis. The selected protein sequences were aligned using MAFFT[125] with the L-INS-i algorithm. From these aligned sequences, a maximum likelihood phylogenetic tree was constructed using IQ-Tree[126] with 1000 ultrafast bootstrap replicates[127]. Automatic model selection identified WAG + R9 as the best-fit model. The resulting phylogeny was visualized using iTOL[128] (Supplementary Fig. 11).

### Downstream analysis of transcript data: Gene Regulatory Network (GRN)

From the time-series transcriptomics data, a Gene Network Reconstruction (GNR) was constructed for each species and treatment individually using Sliding Window Inference for Network Generation (SWING). Log-transformed TPM values were utilized, and the data was filtered for transcripts with significant temporal variation (ANOVA, unadjusted *p*-value < 0.0001) across all timepoints (0, 3, 6, 9, and 25 hours). Edges were computed in SWING using the following parameters: k_min = 1, k_max = 3, window size (w) = 3, method = "RandomForest," and trees = 500. To construct the GNRs, the top 0.1% of scoring edges were retained, consistent with the cut-off that proved effective in comparing streptophyte algal GRNs in light of evolution[14]. For the osmotic stress GNR of each species, common edges between the salt and mannitol time-series networks were retained, except for edges that were also present in the control time-series network. Unfortunately, no edges remained for constructing a network for either *Zygnema* or *Mesotaenium*.

### Long term salt stress

For one *Zygnema* and one *Mesotaenium* plate, cellophane discs containing algae were transferred onto fresh plates with the same cultivation medium, supplemented with 0.15 M NaCl. These plates were then placed in a long-term storage cabinet under constant light conditions (light intensity ≈ 10 μE) and a temperature of 14 °C. After 26 days, we examined the algal morphology using a Zeiss Axioscope 7 light microscope. In *Zygnema*, we observed deterioration of the cell wall, while in *Mesotaenium*, we noted the formation of short filaments (maximum observed length = 10 cells/filament). This prompted us to set up a growth experiment: The strains SAG 12.97 *Mesotaenium endlicherianum* and SAG 698-1b *Zygnema circumcarinatum* were cultivated on solid Warris-H medium under standard culture conditions (20 °C, 30 μE, 14:10 L/D cycle). In the middle of log-phase after 14 days of growth, the algae were resuspended in 2 ml of liquid Warris-H medium containing 0.15 M NaCl. 300 μl of this suspension was placed on Petri dishes (Ø 9 cm) with solid Warris-H medium containing 0.15 M NaCl with and without sterile cellophane disks. The suspension was carefully spread out on the top of the agar surface with sterile Delta™ Disposable Cell Spreader (Merck, Darmstadt, Germany). The algae were grown for eight weeks. The microscopical control was performed every seven days. The Olympus BX-60 microscope (Olympus, Tokyo, Japan) equipped with camera Gryphax Prokyon and the software Gryphax ver. 2.2.0.1234 (both from Jenoptik, Jena, Germany) was used for taking microphotographs. The experiments were performed in three technical replicates.

*Zygnema* bleached out after 48 hours of exposure to the salt stress and was therefore excluded from further experiments. In contrast, *Mesotaenium* showed satisfied growth on medium containing salt. The number of short filaments were measured for all taken images; chains of loosely connected cells were counted as individual cells, while a continuous cell wall spanning multiple cells was required for a structure to be considered as filament.

## Software list

The following software has been used: BLAST[129] (v2.16.0), BUSCO[130] (v5.4.3), dbCAN[131] (v3), FASTQC[132] (v0.11.9), FUNC-E[133] (v2.0.1), ggplot2 (v3.5.1)[134], igraph[135] (v2.0.3), imageJ[136] (v1.54 g), iqtree[126] (v2.1.3), iTOL[137] (v7), InterProScan[123] (v5.72), Kallisto[138] (v0.46.2), MAFFT[125] (v7.304b), Maxqant[139] (v2.4.14.0), MSconvert[140] (v3), MSnbase[141] (v2.28.1), MultiQC[142] (v1.11), mzmine[143] (v4.3.0), NetGPI[144] (v1.1), Perseus[145] (v2.1.1.0 and v2.0.7.0), PTXQC[146] (v1.1.1), Python v3.9.7, Scikit-learn[147] (0.24.2), R 4.3.2, scipy[148] (v1.7.1), Seaborn[149] (v0.13.2), Sleuth[150] (v0.30), SMART[151] (v9.0), Snakemake[152] (v7.7.0), statsmodels 0.12.2, SWING[153], TargetP[154] (v2.0), TransDecoder[155] (v5.7.1), Trinity[156,157] (v2.15.1), trimmomatic[158] (v0.36).

## Reporting summary

Further information on research design is available in the Nature Portfolio Reporting Summary linked to this article.

## Data availability

All RNAseq data generated in this study have been deposited on NCBI under Bioproject ID PRJNA1217932. Micrographs data generated in this study and used for quantifications and additional information have been uploaded on Zenodo under https://doi.org/10.5281/zenodo.14777872 [https://doi.org/10.5281/zenodo.14777872] and https://doi.org/10.5281/zenodo.17651141 [https://doi.org/10.5281/zenodo.17651141]. The mass spectrometry proteomics data generated in this study have been deposited to the ProteomeXchange Consortium via the PRIDE[159] partner repository with the dataset identifier PXD060327. The metabolomics data generated in this study have been deposited to MetaboLights[160] repository with the study identifier MTBLS12267. Source data are provided with this paper.

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

## Acknowledgements

J.d.V. thanks (a) the European Research Council for funding under the European Union's Horizon 2020 research and innovation program (Grant Agreement No. 852725; ERC-StG "TerreStriAL") and (b) the German Research Foundation (DFG) for support through project "SHOAL" (514060973), project "Klebsome" (509535047) and the framework of the Priority Program "MAdLand – Molecular Adaptation to Land: Plant Evolution to Change" (SPP 2237; 440231723 and 528076711), in which

C.F.K. and J.M.S.Z. partake as associate members. C.F.K. and J.M.S.Z. are grateful for support through the IMPRS Genome Science. B.C. and L.P. thank the German Research Foundation (DFG) for support through the framework of the Priority Program "MAdLand – Molecular Adaptation to Land: Plant Evolution to Change" (SPP 2237; 440046237). K.S. is financed by DFG INST 186/1465-1. The HF LC-MS system ultimate 3000-Q Exactive was granted by DFG INST 186/1230-1 FUGG. K.S. is financed by DFG INST 186/1465–1. We extend our heartfelt gratitude to Dr. Ilka Nacif Abreu, Prof. Dr. Ivo Feußner, and Dr. Kirstin Feußner for their expert guidance in metabolomics analysis and their invaluable contribution to performing the metabolomic measurements (facility and equipment supported by the DFG, project number 495720893). We also wish to thank René Heisse for providing exceptional technical assistance during the stress experiments.

## Author contributions

J.d.V. and J.M.S.Z. conceived the project; J.d.V. coordinated the project with J.M.S.Z.; J.d.V. and J.M.S.Z. designed the experiments; J.M.S.Z., L.P., T.D., K.S., C.A.D., C.F.K. and O.V. performed experimental work. T.D. performed the long-term salt experiments. C.A.G. performed cell staining with the Yariv reagent. J.M.S.Z. and J.d.V. designed the computational analysis. J.M.S.Z. carried out computational analysis. J.M.S.Z., K.S., G.H.B., and O.V. performed proteomics. J.M.S.Z. and C.F.K. performed metabolomics. L.P. and B.C. performed AGP and sugar analysis. J.M.S.Z. designed figures. J.M.S.Z. and J.d.V. organized and wrote the final manuscript. All authors commented, discussed, and provided input on the final manuscript.

## Funding

## Competing interests

The authors declare no competing interests.
