## [Transparent Peer Review file · Nature Communications]

Systems acclimation to osmotic stress in zygnematophyte cells

Corresponding Author: Professor Jan de Vries

Version 0:

Reviewer comments:

Reviewer #1

(Remarks to the Author)

This paper compares response to osmotic stress in two forms of zygnematophyte cells in order to provide insight into the adaptation to terrestrial environments. The authors analyze transcriptomic, proteomic, and metabolic data at several timepoints after exposed to osmotic stress in the form of salt and mannitol. Changes in the cell wall were identified and follow up experiments performed to obtain further insight into the cell wall changes. I believe this is an important study of the evolution to land plants. To my knowledge, the methods used were appropriate.

In general, the manuscript is well written, but the figures are complex and need more information in the figure legends to improve understanding. I recommend some minor changes to improve readability of the manuscript:

1. Line 45, "Here the focus was particularly on signaling cascades". This is unclear because signal cascades were not mentioned in the manuscript and a sentence on line 46, "Here, we focused 47 on osmotic stress, the prime challenge on land." details a different focus.
2. In Figure 1 it would be helpful to explicitly mention the meanings of each color, instead of the information only being in one of the images. B and C should be switched in the figure description. Please explain what the P values are corresponding to in the graphs.
3. Line 133 and 135 should say figs 1D, 1E
4. Should Line 123 say 1C only?
5. I could use more information in the figure text on how to interpret 3a and 3b. What is the white line? What is negative rank?
6. Fig 4 needs A and B labeled. Need more information on how to interpret the graphs. What are the red dotted lines?
7. Can you separate figure 5 into A and B for better clarity?
8. Figure 6D bottom, what are the units on the ELISA measurements?
9. Data accessibility: Line 812 Please specify that the RNAseq data has been deposited to NCBI. Line 813 Please clarify how to access micrographs. Please make sure all the data is accessible upon publication. I was unable to find the proteomics and metabolomics data in their respective repositories.
10. The discussion may benefit from a conclusion paragraph integrating a broader discussion on the evolution of algae to the terrestrial habitat and how your results fit into that narrative.

Reviewer #2

(Remarks to the Author)

The study investigated the response of two algae, *Mesotaenium* and *Zygnema*, to osmotic stress. The algae were exposed

to high concentrations of mannitol and NaCl to simulate osmotic stress. Both algae responded to osmotic stress by reducing their water content and undergoing plasmolysis, with *Mesotaenium* showing a more pronounced response. Proteomic and transcriptomic analysis revealed a complex response to osmotic stress, with many proteins and transcripts changing in abundance.

The study found that both algae underwent cell wall remodeling in response to osmotic stress, with the up regulation of cell wall-associated proteins such as XTH and expansins. The cell wall of *Mesotaenium* became more compact and rigid, while that of *Zygnema* became looser and more fragile. The study also found that *Mesotaenium* AGPs were released into the extracellular space in response to osmotic stress. The study highlights the complex response of algae to osmotic stress, with multiple layers of biophysical feedback mechanisms involved, and suggests that the response of algae to osmotic stress is highly conserved across species.

The manuscript presents important data of interest to a broad community, highlighting the conserved background present in extant zygmatophyte algae for abiotic stress resilience. As expected from this lab, the bioinformatic analyses are state of the art and the figures and illustrations are of high quality.

However, I am not entirely convinced that osmotic stress is the best model for abiotic stress during land colonization (as somehow claimed in the introduction). It is easier to do than any desiccation stress, but much milder. That said, I still think this is a very important contribution to the field and should be published. I have a few minor comments that the authors should consider and address prior to publication.

Line 47: Main challenge on land: I doubt that osmotic change is the main challenge on land. It is desiccation or drought, which has many similarities but also clear differences to osmotic stress, especially if you incubate with a salt! See also below.

Line 49: Vascular plants are homoiohydric: Not all, e.g. resurrection plants, and seeds in general are also poikilohydric. Vascular plants have just transferred desiccation tolerance to spores and seeds.

Line 53: Waxy cuticle: As far as I know, this is really rare. I find water conducting tissues and stomata in sporophytes a better example.

Line 55: delves into poikilohyry: Reading this I would expect a desiccation-drought study, not osmolarity, see above!

Line 79: on osmotic stress, thriving in habitats subject to frequent desiccation: Again, you are confusing osmotic stress and desiccation stress. When a ditch in which *Zygnema* thrives dries out, there is a good chance that there will be initial osmotic stress due to increasing osmolarity, but I doubt that there will ever be anything like the 800 mM mannitol you used in your experiments. When the water is gone, the real desiccation stress starts, which will cause much more water loss than osmotic stress. However, osmotic stress can acclimate the alga to desiccation stress. In desiccation you can always reach a YII of zero in PAM, which shows how much stronger the desiccation stress is!

Line 106 to 108: Osmolarity of the media: The total osmolarity is slightly higher because you added 150 mM NaCl or 800 mM mannitol to the growth medium. Do you know the osmolarity of your media? Or did you change the media to 800 mM mannitol so that you have not only osmotic stress but also nutrient starvation!

Line 115: Decreased in both algae: Not true for *Zygnema* and mannitol, if I interpret Fig. 1 A correctly.

Line 118 - 119: never approached zero: confirms that osmotic stress is much milder than desiccation. When you desiccate, you always reach zero!

Line 133: structures induced by mannitol treatment Figure 1D visible tonoplast: I doubt that you can see a tonoplast membrane in a light microscope. Be more precise! Better vacuolated. Many of the LM observations are not significant. And the results are strange. For example, if you compare curved and cupped for *Mesotaenium*, I would have thought that both would be insignificant. Why show all the insignificant results? For *Mesotaenium* 12 out of 16 comparisons are not significant $P = 0.07 - 0.642$! For *Zygnema* it is much better, but you show all the insignificant cell shapes for *Mesotaenium*, but not all for *Zygnema* -

Lane 143: 9600 *Zygnema* transcripts. This is a rather low value for *Zygnema*. The genome paper has 16,617 annotated genes. Very often you have more transcripts than genes in RNASeq. I am not sure if the RNASeq data are reliable for *Zygnema*.

Fig. 2 A: Why are there three PCAs for each species at the protein level? Give time points!!! This is not explained anywhere. It is very difficult to see the differences for the treatments. And I think the differences at the protein level are much smaller than at the RNA level. For the RNA Seq data, the effect arrows do not always show the correct direction of the effect. For example, for the mannitol treatment and *mesotaenium*, the effect is clearly in the direction of the upper left corner, not vertical! Similarly, the effect vectors for *Zygnema* control and Salt are not correct.

Line 145. Why did you assemble de novo? Genomes are available?

Line 164: Difference in effect strength between protein and RNA. I do not see this. Also in the PCA (Fig. 2 A), I see a stronger effect for 25 h (darkest color). The most surprising result for me is that there is no variation in the control for *mesotaenium* and the longest salt treatment is again identical to the control! But these are not even mentioned in the text.

Minor: The legends for the figures should be improved. They should be understandable without text.

Reviewer #3

(Remarks to the Author)

The details surrounding the colonization of land by early embryophytes are poorly understood. These organisms likely inhabited a freshwater environment, perhaps with periodic desiccation, ultimately selecting for adaptations that would lead to the colonization of dry land. Although osmotic stress responses in embryophytes are well studied, such responses are

poorly characterized in zygnematophytes, the closest relatives of land plants. Understanding the modern biology of both the zygnematophytes and embryophytes is crucial to reconstructing the biology of their common ancestor. It has practical implications as well, because plant stress causes substantial losses in agricultural productivity, and these losses are likely to increase as the climate changes. To better understand these stress responses, Zegers et al. characterized the osmotic stress responses of two zygnematophytes, *Mesotaenium endlicherianum* and *Zygnema circumcarinatum*, using transcriptomic, proteomic, and metabolomic techniques. The authors provide evidence that some of the responses to osmotic stress utilized by modern-day embryophytes may also be present in some taxa of zygnematophytes.

The authors performed classic drought stress experiments on these zygnematophytes, echoing work that has been done on embryophytes. Their transcriptomic results show that both zygnematophytes display a rapid response to salt stress with adaptation to .15M NaCl after 25 hours. Mannitol treatment on the other hand showed a stronger response which suggests these organisms have a much more difficult time adapting to the osmotic stress that mannitol causes. However, as the authors state, *Zygnema* has a “temporally dynamic response in the control condition” which makes it difficult to draw any conclusions from that species’ given data.

This is emblematic of the fundamental problem with the manuscript: although the work is of high quality and the data will significantly contribute to the field, some analyses feel unmotivated or out of place. For example, the short discussion of the principal component analysis seemed disconnected from the discussion. What is the significance of movement in this feature space over time? How should this be interpreted? Can the results be extended beyond the two organisms studied? It is also unclear what conclusions, if any, can be obtained from analyzing the correlation between transcript and protein levels. There appears to be a correlation between *Mesotaenium* and the response to salt stress over time but because of the nature of PCA no significant conclusions can be drawn from this. The protein and transcript levels at a given point in time cannot be easily interpreted without knowledge of protein/transcript turnover rates and transcription/translation rates.

Importantly, the discussion of their findings is quite short and cursory, meaning that the manuscript as a whole lacks a coherent message. It may be that this is because very few of the transcripts and proteins found significantly upregulated in the study were identified. The manuscript presents a lot of data, but doesn’t address what it means for early land plant evolution. Similarly, the biological functions of probable embryophyte homologs are discussed, but the authors do not synthesize a clear image of what this means for zygnematophyte biology.

It is an impressive dataset and ultimately will contribute to a better understanding of stress responses in both zygnematophyte algae and in embryophytes, but in its current form seems likely to be of interest only to specialists in the field. They, however, will be very interested.

There are a few typos and grammatical errors here and there, but on the whole the manuscript was carefully prepared. The sentence on lines 273-274 is malformed.

Reviewer #4

(Remarks to the Author)

Reviewer #5

(Remarks to the Author)

Version 1:

Reviewer comments:

Reviewer #1

(Remarks to the Author)

Thank you for addressing my suggestions/concerns. The updated images, legends and conclusion have improved the readability of this article.

Reviewer #2

(Remarks to the Author)

Dear authors, I appreciate the changes you made to the manuscript. As far as I am concerned, the manuscript can be now published.

Reviewer #3

(Remarks to the Author)

I am satisfied by the revisions. The paper has always been interesting, and as revised the motivation behind individual components of the project is clearer, and the implications of the observations are easier to understand. I appreciate the clearer discussion of the relative effects of hypo- and hyper-osmotic stresses and their environmental significance. I feel the manuscript is now ready for publication with no further modification.

Reviewer #5

(Remarks to the Author)

RESPONSE TO REVIEWERS' COMMENTS

Reviewer #1 (Remarks to the Author):

This paper compares response to osmotic stress in two forms of zygnetomophyte cells in order to provide insight into the adaptation to terrestrial environments. The authors analyze transcriptomic, proteomic, and metabolic data at several timepoints after exposed to osmotic stress in the form of salt and manitol. Changes in the cell wall were identified and follow up experiments performed to obtain further insight into the cell wall changes. I believe this is an important study of the evolution to land plants. to my knowledge, the methods used were appropriate.

>>>>AU: Thank you for appreciating our work.

In general, the manuscript is well written, but the figures are complex and need more information in the figure legends to improve understanding. I recommend some minor changes to improve readability of the manuscript:

>>>>AU: We have made sure to address all of the throughout very helpful comments.

1. Line 45, "Here the focus was particularly on signaling cascades". This is unclear because signal cascades were not mentioned in the manuscript and a sentence on line 46, "Here, we focused 47 on osmotic stress, the prime challenge on land." details a different focus.

>>>>AU: The "signaling" focus refers to previous conducted studies and the general osmotic stress response to this study, but we see that using 2 times the word "here" is confusing. We rephrased this in the revised version.

2. In Figure 1 it would be helpful to explicitly mention the meanings of each color, instead of the information only being in one of the images. B and C should be switched in the figure description. Please explain what the P values are corresponding to in the graphs.

>>>>AU: All added and corrected.

3. Line 133 and 135 should say figs 1D, 1E

>>>>AU: Thanks for spotting this. We were indeed not consistent. Now all was changed from "(Fig 1c,d)" to the style "(Figs. 1c, 1d)".

4. Should Line 123 say 1C only?

>>>>AU: Indeed, has been corrected.

5. I could use more information in the figure text on how to interpret 3a and 3b. What is the white line? What is negative rank?

>>>>AU: Good point. Explanation has been added to the figure description.

6. Fig 4 needs A and B labeled. Need more information on how to interpret the graphs. What are the red dotted lines?

>>>>AU: True. We added labels. The red dotted lines are just to make it easier to figure out which term is shown in Zygnema without the reader having to use a ruler. This has been slightly changed in Figure and Figure description to avoid confusion.

7. Can you separate figure 5 into A and B for better clarity?

>>>>AU: Good point; done.

8. Figure 6D bottom, what are the units on the ELISA measurements?

>>>>AU: The units are arbitrary and show the relative value to the highest measured signal for each antibody seperately. We added this in the description for clarification.

9. Data accessibility: Line 812 Please specify that the RNAseq data has been deposited to NCBI. Line 813 Please clarify how to access micrographs. Please make sure all the data is accessible upon publication. I was unable to find the proteomics and metabolomics data in their respective repositories.

>>>>AU: We have added the information that this is on NCBI; regarding all other datasets. Everything is online now.

10. The discussion may benefit from a conclusion paragraph integrating a broader discussion on the evolution of algae to the terrestrial habitat and how your results fit into that narrative.

>>>>AU: True, a small conclusive paragraph has been added.

Reviewer #2 (Remarks to the Author):

The study investigated the response of two algae, *Mesotaenium* and *Zygnema*, to osmotic stress. The algae were exposed to high concentrations of mannitol and NaCl to simulate osmotic stress. Both algae responded to osmotic stress by reducing their water content and undergoing plasmolysis, with *Mesotaenium* showing a more pronounced response. Proteomic and transcriptomic analysis revealed a complex response to osmotic stress, with many proteins and transcripts changing in abundance.

The study found that both algae underwent cell wall remodeling in response to osmotic stress, with the up regulation of cell wall-associated proteins such as XTH and expansins. The cell wall of *Mesotaenium* became more compact and rigid, while that of *Zygnema* became looser and more fragile. The study also found that *Mesotaenium* AGPs were released into the extracellular space in response to osmotic stress. The study highlights the complex response of algae to osmotic stress, with multiple layers of biophysical feedback mechanisms involved, and suggests that the response of algae to osmotic stress is highly conserved across species.

The manuscript presents important data of interest to a broad community, highlighting the conserved background present in extant zygnematophyte algae for abiotic stress resilience. As expected from this lab, the bioinformatic analyses are state of the art and the figures and illustrations are of high quality.

>>>>AU: Thank you for the kind words and appreciating our work.

However, I am not entirely convinced that osmotic stress is the best model for abiotic stress during land colonization (as somehow claimed in the introduction). It is easier to do than any desiccation stress, but much milder. That said, I still think this is a very important contribution to the field and should be published. I have a few minor comments that the authors should consider and address prior to publication.

Line 47: Main challenge on land: I doubt that osmotic change is the main challenge on land. It is desiccation or drought, which has many similarities but also clear differences to osmotic stress, especially if you incubate with a salt! See also below.

>>>>AU: Thank you for this comment — it is a good point. We have clarified that desiccation and drought are a form of osmotic stress: By removing water from the environment (desiccation), the concentrations of osmolytes and other solutes around the cells increase, which leads to hyperosmotic stress. We extensively clarified this in the text. Further, we have created an all-new figure (Fig. 7) that illustrates the cell biological consequences and places them in a phylogenetic context.

Line 49: Vascular plants are homoiohydric: Not all, e.g. resurrection plants, and seeds in general are also poikilohydric. Vascular plants have just transferred desiccation tolerance to spores and seeds.

>>>>AU: Thank you for pointing the case of resurrection plants out. We changed the second paragraph so that it better fits the overall story and omitted poikilohydricity altogether.

Line 53: Waxy cuticle: As far as I know, this is really rare. I find water conducting tissues and stomata in sporophytes a better example.

>>>>AU: We now condensed this while briefly mention all.

Line 55: delves into poikilohyry: Reading this I would expect a desiccation-drought study, not osmolarity, see above!

>>>>AU: Thank you. This issue has been solved in the point above (line 47 and line 49).

Line 79: on osmotic stress, thriving in habitats subject to frequent desiccation: Again, you are confusing osmotic stress and desiccation stress. When a ditch in which *Zygnema* thrives dries out, there is a good chance that there will be initial osmotic stress due to increasing osmolarity, but I doubt that there will ever be anything like the 800 mM mannitol you used in your experiments. When the water is gone, the real desiccation stress starts, which will cause much more water loss than osmotic stress. However, osmotic stress can acclimate the alga to desiccation stress. In desiccation you can always reach a YII of zero in PAM, which shows how much stronger the desiccation stress is!

>>>>AU: Thank you for sparking this important discussion. It was not our intention to intermingle desiccation (complete removal of water, leading to increased extracellular osmolyte concentrations) with osmotic stress (stress due to changes in extracellular osmolyte concentrations). Indeed, as the title of our manuscript indicates, the focus is indeed on acclimation to osmotic stress. However, we see that wording we used could imply that we tried to mimick these extreme circumstances. (which we did not as we liked to stay inside the stress response range of our organisms were we could theoretically obtain a clear transcriptomic and proteomic change.) Hence, this sentences has been adapted.

Line 106 to 108: Osmolarity of the media: The total osmolarity is slightly higher because you added 150 mM NaCl or 800 mM mannitol to the growth medium. Do you know the osmolarity of your media? Or did you change the media to 800 mM mannitol so that you have not only osmotic stress but also nutrient starvation!

>>>>AU: Good point – we did indeed calculate the osmolarity of the medium, albeit neglecting the contribution of agar, which consist of long polymers and therefor contributes close to 0 to the osmolarity. The osmolarity of the standard C medium is 0.015 Osm/L and for WHM medium the osmolarity is 0.018 Osm/L. Adding NaCl raises the osmolarities to 0.315 Osm/L and 0.318 Osm/L, whereas adding the mannitol raises the osmolarities to 0.815 Osm/L and 0.818 Osm/L. The numbers were added to the method section.

We hope that it is now more precisely written that the media was (of course) not replaced with just mannitol/salt medium, but it was supplemented to the existing media (all in the method section). The final concentrations of micronutrients remained equal to prevent nutrient stress.

Line 115: Decreased in both algae: Not true for Zygnema and mannitol, if I interpret Fig. 1 A correctly.

>>>>AU: Error well spot, it should say 6 hours our, not 3 hours. Has been adapted.

Line 118 - 119: never approached zero: confirms that osmotic stress is much milder than desiccation. When you desiccate, you always reach zero!

>>>>AU: Correct. The osmotic stress applied here is indeed milder than full desiccation, which is also a form of osmotic stress in which the osmolarity increases towards indefinitely and due to the absence of water poses a much larger risk of cell wall collapse than in our experiments.

Line 133: structures induced by mannitol treatment Figure 1D visible tonoplast: I doubt that you can see a tonoplast membrane in a light microscope. Be more precise! Better vacuolated. Many of the LM observations are not significant. And the results are strange. For example, if you compare curved and cupped for Mesotaenium, I would have thought that both would be insignificant. Why show all the insignificant results? For Mesotaenium 12 out of 16 comparisons are not significant $P = 0.07 - 0.642!$ For Zygnema it is much better, but you show all the insignificant cell shapes for Mesotaenium, but not all for Zygnema

>>>>AU: With regard to the tonoplast: to distinguish tonoplast from another membrane with light microscope is indeed hard and you cannot be certain. Hence we added right above it in the figure "presumably, see method section for detailed description". We enlarged the text in the figure a bit so it would not be missed. Also, the algae should also have vacuole in control conditions, so vacuolated (=having a vacuole) we think is not a better term. For transparency, we showed all microscopic features that were statistically analysed. For Mesotaenium, we thought we saw or suspected more colouration and more lipid droplets within the stress treatments, but after counting it turned out that it was not the case (or at least not significant). Negative results are also results. We now highlighted all P-values below 0.05 for better visualisation.

Line 143: 9600 Zygnema transcripts. This is a rather low value for Zygnema. The genome paper has 16,617 annotated genes. Very often you have more transcripts than genes in RNASeq. I am not sure if the RNASeq data are reliable for Zygnema.

>>>>AU: We compared our numbers with the very recent manuscript of Rieseberg et al and our number of genes (they analysed on gene-level) were very similar (9445 genes in the other work and 9620 genes in our case), so even slightly higher in our case. One should take into account that the genome paper took a complementing approach of very diverse conditions (our focus was „just“ on osmotic stress, whereas in the genome paper various stressors and the diurnal cycle were sampled, where the night alone causes a complete change in the active gene set), which often lead to the detection of more genes as some are only expressed under certain conditions. On top of that, we filtered extremely low abundant transcript over all conditions (around 1500 transcripts, who can hinder the DEanalysis with relative high LOG2foldchanges when absolute the change is minimal) out and Zygnema has for the majority of genes only a single highly abundant transcript. We added 1 sentence to the manuscript were the numbers of Rieseberg et al are being compared.

Fig. 2 A: Why are there three PCAs for each species at the protein level?

>>>>AU: Since PC1, PC2, and PC3 combined best explain the variation in the data, we provided all to show a more complete picture of separation of the data points.

Give time points!!! This is not explained anywhere. It is very difficult to see the differences for the treatments.

>>>>AU: While it was given in the legend at the bottom, it can indeed be made clearer. We thus adapted it.

And I think the differences at the protein level are much smaller than at the RNA level.

>>>>AU: Correct, which is why this combined approach is very valuable in our opinion.

For the RNA Seq data, the effect arrows do not always show the correct direction of the effect. For example, for the mannitol treatment and mesotaenium, the effect is clearly in the direction of the upper left corner, not vertical! Similarly, the effect vectors for Zygnema control and Salt are not correct.

>>>>AU: The vectors have been slightly corrected

Line 145. Why did you assemble de novo? Genomes are available?

>>>>AU: We used both the genome annotation that are available, but as a complementing approach we also added de novo assembled information. Both with these non-model systems (or budding models) and even established model species, gene models get frequently updated. The genome of Mesotaenium is a point in case (V2 in Dadras et al. 2023, owing to the increase in RNAseq coverage). Both for Mesotaenium and for Zygnema, we found proteins that were not in the established database. Doing the transcriptome assembly was in hindsight not strictly necessary for this work, but a nice addition – we considered it better to be as comprehensive and careful as possible rather than omitting this to save time and work.

Line 164: Difference in effect strength between protein and RNA. I do not see this. Also in the PCA (Fig. 2 A), I see a stronger effect for 25 h (darkest color).

>>>>AU: While we do not see this stronger effect at 25 hours, we agree that this effect is not that clear for each condition: for salt stress it is, but for mannitol it is less pronounced. Hence, we amended this part of the text.

The most surprising result for me is that there is no variation in the control for mesotaenium and the longest salt treatment is again identical to the control! But these are not even mentioned in the text.

>>>>AU: We briefly mention the lack of changes in control treatment (which is indeed a very nice circumstance for DE analyses) in Mesotaenium later when talking about the clustering. We will mention it as well when describing the PCA+DEanalysis.

Minor: The legends for the figures should be improved. They should be understandable without text.

>>>>AU: Good point. We took several measures: (i) Fig2b+Fig2c legend improved. (ii) Fig3b: axis improved. (iii) Fig5a: minor legend change. (iv) Fig6a: minor legend visualisation improvement. (v) Fig6b: addition of NaCl labels. (vi) Fig6F: addition of "Osmo-responsive subfam."

Reviewer #3 (Remarks to the Author):

The details surrounding the colonization of land by early embryophytes are poorly understood. These organisms likely inhabited a freshwater environment, perhaps with periodic desiccation, ultimately selecting for adaptations that would lead to the colonization of dry land. Although osmotic stress responses in embryophytes are well studied, such responses are poorly characterized in zygmatophytes, the closest relatives of land plants. Understanding the modern biology of both the zygmatophytes and embryophytes is crucial to reconstructing the biology of their common ancestor. It has practical implications as well, because plant stress causes substantial losses in agricultural productivity, and these losses are likely to increase as the climate changes. To better understand these stress responses, Zegers et al. characterized the osmotic stress responses of two zygmatophytes, *Mesotaenium endlicherianum* and *Zygnema circumcarinatum*, using transcriptomic, proteomic, and metabolomic techniques. The authors provide evidence that some of the responses to osmotic stress utilized by modern-day embryophytes may also be present in some taxa of zygmatophytes. The authors performed classic drought stress experiments on these zygmatophytes, echoing work that has been done on embryophytes. Their transcriptomic results show that both zygmatophytes display a rapid response to salt stress with adaptation to .15M NaCl after 25 hours. Mannitol treatment on the other hand showed a stronger response which suggests these organisms have a much more difficult time adapting to the osmotic stress that mannitol causes. However, as the authors state, *Zygnema* has a "temporally dynamic response in the control condition" which makes it difficult to draw any conclusions from that species' given data.

This is emblematic of the fundamental problem with the manuscript: although the work is of high quality and the data will significantly contribute to the field, some analyses feel unmotivated or out of place.

>>>>AU: Thank you for the assessment of our manuscript which sparked many important discussion points amongst us. We have taken all your feedback into account and improved the discussions, phrasing, and framing in our manuscript. Regarding the temporal dynamic in *Zygnema*: we explain in the manuscript what the biological meaning of this could – but more importantly, we have taken this in our clustering and functional enrichment into account, which is what all major conclusions are based on; more below.

For example, the short discussion of the principal component analysis seemed disconnected from the discussion. What is the significance of movement in this feature space over time? How should this be interpreted?

>>>>AU: This is the response and acclimation process over time. We expanded the explanation on this.

Can the results be extended beyond the two organisms studied?

>>>>AU: Indeed, we think they can. We have pronouncedly extended the discussion on this aspect

It is also unclear what conclusions, if any, can be obtained from analyzing the correlation between transcript and protein levels.

>>>>AU: If the reviewer is referring to fig. 3, the main aim of this is to first show a bird's eye view of the correlation. We then move to talking about specific proteins/protein coding genes that show consistent patterns on RNA and protein level, and we highlight also proteins/transcript where the opposite is the case.

There appears to be a correlation between *Mesotaenium* and the response to salt stress over time but because of the nature of PCA no significant conclusions can be drawn from this

>>>>AU: Both for salt and mannitol, the PCA shows clear clustering of all control treatments and separation of the treatments in *Mesotaenium*, not hinting towards possible underlying mechanisms that play a more prominent role than the applied osmotic stress. The main aim of the PCA is to provide an overview. Our inferences of the mechanisms are what follows next. We have made sure to provide a better red thread for the section on the PCA and the correlation to highlight this.

The protein and transcript levels at a given point in time cannot be easily interpreted without knowledge of protein/transcript turnover rates and transcription/translation rates.

>>>>AU: We mitigate by performing differential analysis that build on the correct statistics (always comparing to a control or another timepoint) and not just looking at the raw abundances. The only case where raw abundances were used was for the absolute abundance analyses, where the point is exactly the conclusion that a straightforward correlation is poor. We thus turned in all the remaining – and major conclusion determining – parts of the paper to the differential analyses as also outlined in the main text. This also applies to the clustering, and subsequent functional enrichment – all of which is always compared to t0 and proper controls.

Importantly, the discussion of their findings is quite short and cursory, meaning that the manuscript as a whole lacks a coherent message. It may be that this is because very few of the transcripts and proteins found significantly upregulated in the study were identified. The manuscript presents a lot of data, but doesn't address what it means for early land plant evolution. Similarly, the biological functions of probable embryophyte homologs are discussed, but the authors do not synthesize a clear image of what this means for zygmatophyte biology.

>>>>AU: Good point, which also the other reviewers noticed. Due to the initial space limitation we tried to keep this short. We expanded the discussion with a concluding paragraph.

It is an impressive dataset and ultimately will contribute to a better understanding of stress responses in both zygmatophyte algae and in embryophytes, but in its current form seems likely to be of interest only to specialists in the field. They, however, will be very interested.

>>>>AU: We thank the reviewer for appreciating the scope of our work.

There are a few typos and grammatical errors here and there, but on the whole the manuscript was carefully prepared. The sentence on lines 273-274 is malformed

>>>>AU: Corrected.

Reviewer #4 (Remarks to the Author):

>>>>AU: Thank you for contributing to the constructive feedback.

Reviewer #5 (Remarks to the Author):

>>>>AU: Thank you for contributing to the constructive feedback.